# The CHRONOS mission: Capability for sub-hourly synoptic observations of carbon monoxide and methane to quantify emissions and transport of air pollution

David P. Edwards[1], Helen M. Worden[1], Doreen Neil[2], Gene Francis[1], Tim Valle[3], and Avelino F. Arellano, Jr.[4]

[1]National Center for Atmospheric Research (NCAR), Boulder, CO, USA
[2]NASA Langley Research Center, Hampton, VA, USA
[3]Ball Aerospace, Boulder, CO, USA
[4]University of Arizona, Tucson, AZ, USA

*Correspondence to:* D. P. Edwards (edwards@ucar.edu)

**Abstract.** The CHRONOS space mission concept provides time-resolved abundance for emissions and transport studies of the highly variable and highly uncertain air pollutants carbon monoxide and methane, with sub-hourly revisit rate at fine (~ 4 km) horizontal spatial resolution across a North American domain. CHRONOS can provide complete synoptic air pollution maps ("snapshots") of the continental domain with fewer than 10 minutes of observations. This rapid mapping enables visualization of air pollution transport simultaneously across the entire continent and enables a sentinel-like capability for monitoring evolving, or unanticipated, air pollution sources in multiple locations at the same time with high temporal resolution. CHRONOS uses a compact imaging gas filter correlation radiometer for these observations, with heritage from more than 17 years of scientific data and algorithm advances by the science teams for the MOPITT instrument on NASA's Terra spacecraft in low Earth orbit. To achieve continental-scale sub-hourly sampling, the CHRONOS mission would be conducted from geostationary orbit, with the instrument hosted on a communications or meteorological platform. CHRONOS observations would contribute to an integrated observing system for atmospheric composition using surface, suborbital and satellite data with atmospheric chemistry models, as defined by the Committee on Earth Observing Satellites. Addressing the U.S. National Academy's 2007 Decadal Survey direction to characterize diurnal changes in tropospheric composition, CHRONOS observations would find direct societal applications for air quality management and forecasting to protect public health.

## 1 Introduction

For the end of the current decade, geostationary Earth orbit (GEO) satellite missions for atmospheric composition are planned over North America, East Asia and Europe, with additional missions in formulation or proposed. Together, these present the possibility of a constellation of GEO platforms to achieve continuous, time-resolved, high-density, observations of continental domains for mapping pollutant sources and variability on diurnal and local scales with near-hemispheric coverage (CEOS, 2011). In addition to NASA's TEMPO mission (Zoogman, 2017), the ESA/EUMETSAT Sentinel 4 mission over Europe (GMES-GAS, 2009) and the Korean KARI MP-GEOSAT/GEMS mission over Asia (Lee et al., 2010), will provide data products for ozone ($O_3$), nitrogen dioxide ($NO_2$), sulfur dioxide ($SO_2$), formaldehyde (HCHO) and aerosol optical depth (AOD) several times per day with smaller than 10 km x 10 km spatial footprints. While these planned GEO measurements will provide new information on the diurnal evolution of emissions and chemical transformation of some important pollutants, they are missing observations of methane ($CH_4$) and carbon monoxide (CO). As identified in CEOS (2011), these gases play key roles in atmospheric chemistry, air quality and climate.

The planned GEO constellation will be further enhanced by current and upcoming low Earth orbit (LEO) missions with atmospheric composition measurement capability. These missions include OMI (Ozone Monitoring Instrument, Levelt et al., 2006), IASI (Infrared Atmospheric Sounding Interferometer, Clerbaux et al., 2009), CrIS (Cross-track Infrared Sounder, Gambacorta et al., 2014), OMPS (Ozone Mapping Profiler Suite, Flynn et al., 2014), and the ESA Sentinel-5 precursor mission, TROPOMI (Veefkind et al., 2012). The LEO assets allow for a transfer-standard between the GEO missions, filling gaps in the spatial coverage, enabling cross-calibration and validation, and potentially, combined data products. Such an integrated global observing system for atmospheric composition is key to abatement strategies for air quality as prescribed in international protocols and conventions (e.g., IGACO, 2004).

Pollution affecting air quality is a complex mixture of many compounds that was designated a Group 1 carcinogen by the World Health Organization (WHO) (Loomis et al., 2013) amidst rising concerns about increased mortality and economic costs. Outdoor air pollution causes pulmonary and cardiovascular diseases, lung cancer, and premature birth (Brunekreef and Holgate, 2002; Turner et al., 2015; Fann et al., 2012, Malley et al., 2017). Despite improvements in U.S. air quality

in recent decades, present-day levels of air pollution are estimated to decrease average life expectancy by 0.7 years and contribute to 10% of the total deaths in highly polluted areas such as Los Angeles (Fann et al., 2012). In 2010, over 3% of U.S. preterm births were attributed to air pollution at an estimated cost exceeding $5 billion (Trasande et al., 2016). To address the causes of air pollution effectively, decision makers need comprehensive measurements to quantify the

full suite of pollutants, including $CH_4$ and CO, emitted from industrial, transport and energy sectors, as well as natural sources. CO, which allows detection of combustion-related emissions, serves as the reference for the emissions of many difficult-to-measure pollutants that impact air quality and climate. Wildfires, which emit both CO and $CH_4$, are a particular concern in the Western U.S. (Abatzoglou and Williams, 2016), where burn areas have increased by a factor of 6

since 1970, with severe economic impacts (Westerling et al., 2006). CO and $CH_4$ emissions also have significant consequences for climate change, especially considering $CH_4$ pollution due to recent large increases in natural gas production (Pétron et al., 2012; Miller et al., 2013) and potential new $CH_4$ releases from thawing permafrost (Ciais, 2013).

After air pollutants are emitted, they are transported vertically and horizontally in the atmosphere

and can have a significant impact on local air quality and human health at locations near the sources and also downwind. Distinguishing the relative contributions of local and non-local pollution sources has emerged as a fundamental challenge for air quality management in the U.S. (NRC, 2004). Because CO has a medium lifetime (weeks to months), it can be transported globally, but does not become evenly mixed in the troposphere. This moderate lifetime makes CO an ideal tracer

of combustion-related air pollution (e.g., Edwards et al., 2004; 2006).

The CHRONOS mission is motivated by these fundamental questions regarding the emissions and transport of air pollutants. The CHRONOS gas filter correlation radiometry (GFCR) measurement technique for multi-spectral CO builds on 17 years of observations from the NASA Terra satellite Measurements of Pollution in the Troposphere (MOPITT) instrument (Drummond et al., 2010, H.

M. Worden et al., 2013), in addition to experience in LEO column $CH_4$ retrievals from SCIAMACHY (Frankenberg et al., 2005; 2011) and GOSAT (Morino et al., 2011; Schepers et al., 2012). The CHRONOS temporal resolution (sub-hourly), and spatial resolution (nominally 4 km × 4 km at the domain center), are required to capture the near surface trace gas variability, as concluded by modeling and data studies performed by the Geostationary Coastal & Air Pollution

Events (GEO-CAPE) (Fishman et al., 2012) science team in response to the first Decadal Survey

for Earth Science and Applications (NRC, 2007). For $CH_4$, the spatially and temporally dense CHRONOS measurements over the entire continental U.S. measurement domain would address the need for consistent assessments of $CH_4$ emissions at decision-relevant scales. For CO, proven multispectral retrieval techniques (Worden et al., 2010) increase the information on CO vertical distribution, and can identify vertical transport from one observation to the next. Thus, CHRONOS is capable of tracking pollutants from the surface where they are emitted, to where they degrade downwind air quality.

This paper describes the CHRONOS science, measurement technique, expected performance (precision and accuracy), retrieval vertical sensitivity and observing strategy. We then show how CHRONOS would complement observations from other current and planned satellite instruments, and conclude with a summary of CHRONOS features and advantages.

## 2 CHRONOS Science

### 2.1 CHRONOS Sub-hourly Synoptic Measurements with High Spatial Resolution

Advances in tropospheric remote sensing from LEO over the past decade have shown the potential of satellites to quantify the sources, transport and distributions of the gases important for air quality and climate (NRC, 2007; Simmons et al., 2016). LEO data provide valuable knowledge on continental to global-scale pollution, but their spatial and temporal resolution, sparseness of coverage, and often large uncertainties for individual trace gas observations, have so far limited their use in understanding air pollution sources and distributions on local to regional spatial scales (Figures 1 and 2).

The importance of sub-hourly time resolution for capturing the diurnal evolution of pollution transport is shown in Figure 1, which compares current MOPITT measurement sampling to that which would be obtained from CHRONOS over the continental U.S.

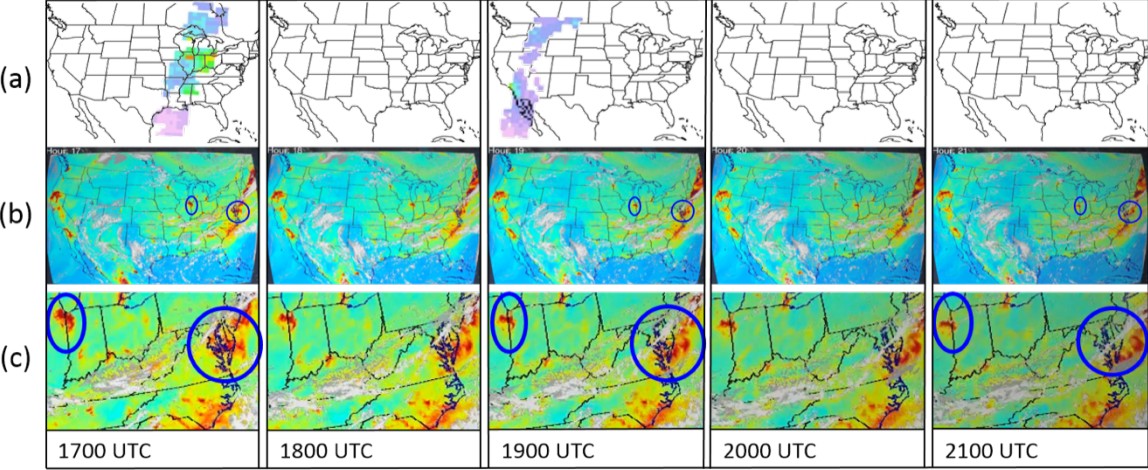

**Figure 1:** Comparison of MOPITT and simulated CHRONOS spatial and temporal coverage over a 5-hour period on Tuesday, August 1, 2006. Top panels (a) show MOPITT retrievals of near-surface CO for each hour, with pink colors indicating low CO (~ 60 ppbV) and green to red indicating higher values (200 – 300 ppbV). No MOPITT data were available at 18:00, 20:00 and 21:00 hours. Middle panels (b) show simulated CHRONOS observations using WRF-Chem (Grell et al., 2005) at 4 km horizontal resolution driven by analyzed meteorology (Barth et al., 2012) for the same date. Here blue colors indicate low CO (~60 ppbV), red colors indicate high CO (~300 ppbV) and light greys indicate clouds. Bottom panels (c) are a magnified view of the simulated CHRONOS observations. Circled areas provide examples of changes in CO concentrations over the 5-hour period with pollution from Chicago moving to the west and clouds moving east over the Washington DC area.

Understanding the rapidly changing tropospheric state and critical processes that are episodic or have diurnal timescales, such as traffic emissions, forest fire intensity, meteorology and changes in the planetary boundary layer (PBL) height, requires temporal resolution that is better than once a day (Fishman et al., 2012). Accurate prediction of air quality requires an observing framework for atmospheric composition similar to that for weather forecasting, where instruments in GEO are essential components of the integrated observing system and complement existing LEO, suborbital, and surface assets and modeling capability. As such, CHRONOS in GEO addresses the

need for sub-hourly vertical and horizontal transport information for "chemical weather" prediction.

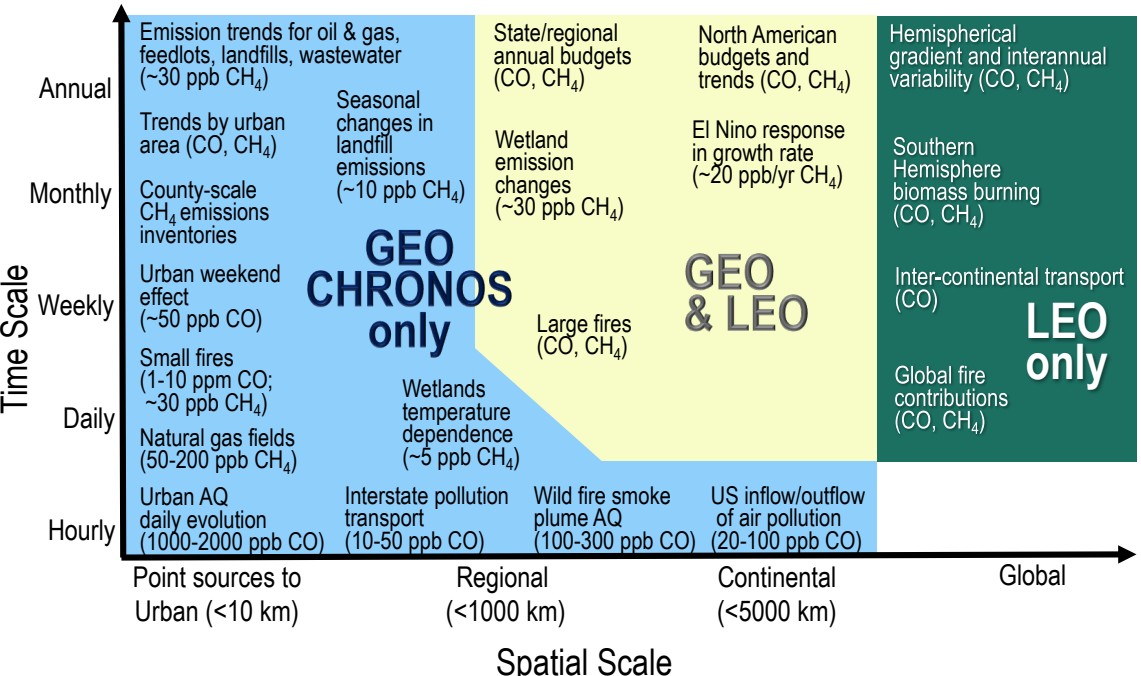

**Figure 2:** CHRONOS' sub-hourly observations would provide access to the fine temporal and fine spatial scales of $CH_4$ and CO processes for understanding the emissions and transport of air pollution for air quality, climate, and energy management applications. Estimated abundances are for process contributions above background levels.

**2.2 The CHRONOS Science Objectives**

CHRONOS focuses on two interrelated science objectives: emissions of highly variable and poorly quantified air pollutants, and air pollution transport across North America. Significant scientific advances in understanding these air pollutant emissions and transport processes are expected to lead to improvements in chemical transport model predictability on both regional and global scales.

**Objective 1 – Emissions:** *Quantify the temporal and spatial variations of $CH_4$ and CO emissions for air quality, climate, and energy decision making.*

Large uncertainties and conflicting estimates exist in current CO and $CH_4$ emissions. Aircraft data show National Emissions Inventory (NEI) CO emissions are too high by a factor of 3 in the

summer (Hudman et al., 2008; Miller et al., 2008). Satellite data, including MOPITT, indicate large seasonal changes in CO emissions with a maximum in winter and minimum in summer (Kopacz et al., 2010), that are currently absent from the NEI, and show fire emissions that are too low by as much as 30% (Pechony et al., 2013). Measurements of $CH_4$ from surface and aircraft observations imply that the EPA 2009 emission inventory is too low, by about a factor of 2, due to large uncertainties from fossil fuel production (coal and natural gas fields, especially in the Western States and Canadian tar sands), transportation, agriculture, wetlands, and thawing permafrost in Canada (Katzenstein et al., 2003; Xiao et al., 2008; Kort et al., 2008; Pétron et al., 2012; Miller et al., 2013; Pechony et al., 2013; Schwietzke et al., 2016). In particular, for natural gas production, recent studies present conflicting results. Karion et al. (2013) showed between 6 and 12% $CH_4$ leakage from gas and oil production fields in Unitah County, Utah, using airborne measurements, while Allen et al. (2013) found less than 1% leakage at 190 U.S. natural gas sites using emissions activity estimates. Recent research identified sensor issues in the surface measurements used by natural gas companies often causing under-estimated emissions (Howard et al., 2015). These observational inconsistencies can be resolved by comprehensive measurements that are temporally and spatially dense.

CO observations also serve as proxy for other pollutant emissions. Emissions of other combustion pollutants that are important to air quality and climate are frequently correlated with CO emissions, including other ozone and aerosol precursors (Edwards et al., 2004; Massie et al., 2006; Zhang et al., 2008; Bian et al., 2010). As a result, the emission inputs to chemical transport models for many combustion-related species are specified by ratios referenced to CO. CO serves as a proxy for anthropogenic carbon dioxide ($CO_2$) (Palmer et al., 2006; Worden et al., 2012; Silva et al., 2013) and black carbon (BC) sources (Arellano et al., 2010). $CH_4$ correlations with CO distinguish $CH_4$ from fires (J. Worden et al., 2013). Assimilation of CHRONOS data into regional scale chemical transport models would leverage inter-species constraints to allow the emissions and distributions of correlated species to be inferred using CHRONOS measurements (e.g., Gaubert et al., 2016).

**Objective 2 – Transport:** *Track rapidly changing vertical and horizontal atmospheric pollution transport to determine near-surface air quality at urban to continental spatial scales, and at diurnal to monthly temporal scales.*

Source attribution for local and transported pollution is an important step toward attaining air quality standards (NRC, 2004). Understanding the production of air pollution requires knowledge of ozone and aerosol precursor emissions (CO and $CH_4$ among them), and the transport of both precursors and other air quality pollutants (for example, using CO as a tracer). Air pollution crosses international and state boundaries to impact downwind cities, national parks, and wilderness areas.

The Cross-State Air Pollution Rule (U.S. EPA, 2011) requires 23 states to reduce emissions in order to meet air quality standards in downwind states. Considerable international efforts are directed toward understanding intercontinental transport of air pollution (Galmarini et al., 2017). CHRONOS' multispectral retrievals of CO would provide the vertical sensitivity to determine transport out of the PBL, into the free troposphere, and the vertical descent back to the surface at

some distance downwind. This new CHRONOS information would allow state and local air quality managers to quantify interstate pollution, along with intermittent sources such as fires that affect the ability of urban areas to meet air quality standards.

The time and space scales of CHRONOS measurements are designed to be similar to the scales of models for regional air quality applications, leading to improvements in process representation.

From observing system simulation experiments (OSSEs), we have demonstrated that data assimilation of simulated CHRONOS multispectral observations of CO significantly improves comparisons with the "true" surface CO values at EPA surface monitoring sites (Edwards et al., 2009). Sub-hourly measurements of CO throughout the troposphere would allow for more frequent data assimilation updates than is currently possible, which, along with increased accuracy in

surface CO knowledge, would dramatically improve the skill for air quality prediction. OSSEs also demonstrate that CHRONOS' CO measurements augment TEMPO's ozone measurement capability through joint ozone-CO data assimilation (Zoogman et al., 2014).

### 2.3 CHRONOS Measurements of $CH_4$ and CO

More than half of $CH_4$ emissions are anthropogenic, with contributions from fossil-fuel

production, animal husbandry and waste management, while wetlands are the primary natural source (Bergamaschi et al., 2009). $CH_4$ has an atmospheric lifetime of 8–10 years, and exerts 86 times the global warming potential of $CO_2$ emissions on a 20-year timeframe (Myhre et al., 2013). The U.S. is presently the world's largest producer of natural gas (Breul et al., 2013). Production has increased 20% since 2008, with a corresponding need to quantify how much $CH_4$ is released

during extraction. Furthermore, $CH_4$ has an impact on air quality as a precursor to tropospheric ozone and aerosols through changes in hydroxyl radical (OH) (Shindell et al., 2009). $CH_4$ thus plays a pivotal role in both air quality and climate, and co-benefits to both air quality and climate may arise from reducing $CH_4$ emissions (West et al., 2006; Shindell et al., 2009; UNEP, 2011; Schneising et al., 2014). CHRONOS' frequent (sub-hourly) $CH_4$ observations would provide the information needed to resolve discrepancies in $CH_4$ emissions at the county, decision-making, scale.

Dense data sampling improves the capability for constraining model emissions (e.g., Bousserez et al., 2016; Wecht et al., 2014 a). Figure 3 shows a grid representing the CHRONOS spatial resolution overlaid on aircraft measurements taken during the FRAPPE-DISCOVER-AQ field campaign (Pfister et al., 2017) in the Colorado Front Range on Aug. 2, 2014. This indicates high $CH_4$ in areas of extensive oil and gas extraction and feedlot operations in Colorado (Greeley and Platteville), as compared to other urban and rural locations. By comparison, $CH_4$ concentrations during the 2015 Aliso Canyon leak (Conley et al., 2016), over the Los Angeles basin were an order of magnitude higher than these Colorado oil and gas concentrations, and thus could have been quantified from space using CHRONOS $CH_4$ observations, had they been available. Recent studies have demonstrated the potential for using CO and $CH_4$ satellite data to constrain sources using adjoint and other inversion models (Bergamaschi et al., 2007; 2009; Meirink et al., 2008; Kopacz et al., 2009; 2010; Fortems-Cheiney et al., 2011; Pechony et al., 2013; Wecht et al., 2014b; Turner et al., 2015; Jacob et al., 2016). These studies also show that present ability to optimize emission estimates is limited by the sparse sampling of present measurements. CHRONOS would provide the data density and near-surface abundance information that are needed in adjoint inversions for CO and $CH_4$ emissions estimates with the spatial and temporal resolution necessary to understand emission inventory errors.

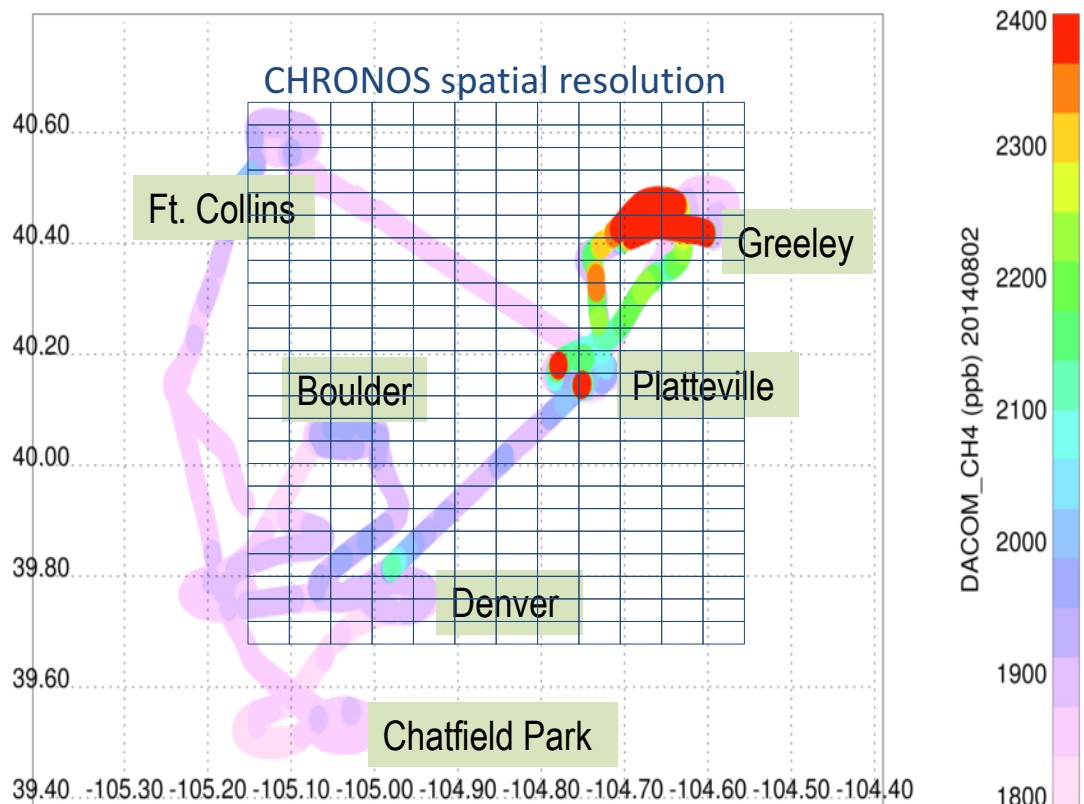

**Figure 3:** Aircraft in situ measurements of CH$_4$ from the FRAPPE-DISCOVER-AQ in the Colorado Front Range on Aug. 2, 2014. Vertical profiles were measured over cities, identified by spiral flight tracks (each spiral has ~10 km radius), where the highest values of CH$_4$ are plotted last. Total column CH$_4$ computed from the vertical profiles is different by 4.9% between Greeley (oil/gas and feedlot operations) and Ft. Collins (urban). CHRONOS pixel spatial resolution is indicated by the overlaid grid, illustrating that CHRONOS column measurements would have the spatial resolution and precision to distinguish sub-hourly differences in county-scale CH$_4$ abundances. Note that grid in this figure only indicates pixel resolution and not the observational domain; CHRONOS images the entire North American field of view in each 60 msec frame. DACOM (differential absorption carbon monoxide monitor) CH$_4$ data are courtesy of Glenn Diskin, NASA.

Nine months before the U.S. Environmental Protection Agency was founded, air quality criteria were established for CO (U.S., 1970) to protect public health in compliance with the 1967

amendments (Public Law 90-148) to the Clean Air Act of 1963 (Public Law 88-206). CO is produced by combustion processes, including transportation, manufacturing, agricultural burning, and wildfires, and by hydrocarbon oxidation. CO participates in the formation of ground level ozone; and, as the dominant sink for the main tropospheric oxidant, OH, CO plays a central role in determining the ability of the atmosphere to cleanse itself of pollutants (e.g., Holloway et al., 2000) and thus affects the lifetime of $CH_4$ (Myhre et al., 2013). The CO lifetime of ~2 months provides time for CO to be transported globally, yet is sufficiently short to show large contrasts between polluted air and the background atmosphere (Edwards et al., 2004). For these reasons, CO is one of the few mission-critical measurements in all aircraft campaigns of the NASA Global Tropospheric Chemistry Program (Fisher et al., 2010) and similar regional air pollution studies. CHRONOS would use the CO multispectral retrieval created by the MOPITT team providing enhanced sensitivity to near-surface CO concentrations (Worden et al., 2010; Deeter et al., 2013). This allows CO plumes near the surface to be distinguished from plumes in the free troposphere to quantify how sources of CO impact downwind regions (Huang et al., 2013). This approach is discussed in Section 5.

**3 The Gas Correlation Filter Radiometry (GCFR) Measurement technique**

**3.1 GCFR Concepts**

Gas filter correlation radiometry features extremely high spectral selectivity combined with high throughput to enable precise measurements of atmospheric trace constituents such as $CH_4$ and CO. GCFR (Acton et al., 1973, Ludwig et al., 1973, Tolton and Drummond, 1997) has been used for satellite remote sensing on Space Shuttle/MAPS (Reichle et al., 1999), UARS/ISAMS and HALOE (Rodgers et al., 1996; Russell et al., 1993), and Terra/MOPITT (Edwards et al., 1999; Drummond et al., 2010). The pioneering MAPS instrument used two detectors with careful electronic balancing on its four Space Shuttle flights to measure CO, and MOPITT uses length and pressure modulation of a single cell, rather than separate gas and vacuum cells, for its successful observations during more than 17 years in LEO. Correlation radiometers have thus proven rugged and reliable in space. The first Decadal Survey for Earth Science and Applications recommended "an IR correlation radiometer for CO mapping" and also stated that the "Combination of the near-

IR and thermal-IR data will describe vertical CO, an excellent tracer of long-range transport of pollution (NRC, 2007)."

The GFCR technique is based on the concept that the near ideal filter for the spectral signal from a particular molecule comes from the molecule itself. The effective spectral resolution of the GFCR response function (Edwards et al., 1999, figure 3) matches the pressure-broadened Lorentz full-width-half-maximum (FWHM) for weak-absorption lines (Beer, 1992), and ranges from 0.08 $cm^{-1}$ to 0.16 $cm^{-1}$ for 200 hPa to 800 hPa GFCR gas cells (Pan et al., 1995). This optimal spectral resolution for measuring tropospheric trace gas absorption and for probing the spectral line profile to obtain information on the trace gas atmospheric vertical distribution is difficult to achieve for most spectrometers without sacrificing signal amplitude (grating spectrometers) or increasing noise (Fourier transform spectrometers). The limitation for the GFCR technique is that atmospheric retrievals are made only for those gases contained within the cells of the instrument. However, for observations of CO and $CH_4$ from GEO (50 times farther from Earth than LEO), the advantages of both high effective spectral resolution and high throughput provided by CHRONOS's gas filter correlation radiometry make for a particularly robust measurement approach.

In the GFCR technique, shown schematically in Figure 4, the top-of-atmosphere (TOA) spectral radiance from each observed field of view (FOV) passes through an instrument cell containing the same gas as the atmospheric target gas being measured, either CO or $CH_4$ in the case of CHRONOS. The instrument cell uses the gas of interest as a highly selective filter to match narrow spectral features in the atmosphere. With known gas cell dimensions, gas content, temperature and pressure, this technique provides nearly perfect spectral knowledge. The GFCR method efficiently filters the target gas information from surrounding spectral interference, while simultaneously measuring and integrating the target spectra across the selected spectral bandpass, delivering a spectral response function that can be accurately calibrated because it is defined by the cell gas absorption. For these reasons, thorough GFCR instrument characterization is needed prior to launch, along with on-orbit radiometric calibration and measurements of cell parameters (Neil et al., 2010).

Idealized implementation of gas filter correlation radiometry requires viewing the same scene through the same optics with the same detector for each of two gas cells (one containing the gas

of interest and the other containing a vacuum (or a gas with no spectral signature in the selected spectral region). The goal is that the ratio of the spectral radiance viewed through the two cells is only a function of the target gas. Spatial misalignment of the two measurements could result in changes in the viewed surface reflectivity, and thus radiance changes in gas-vacuum cell difference. Temporal offsets could result in different atmospheric paths being captured because of target gas or cloud movement through the field of view. Changes in the instrument function between gas and vacuum views (different optics or detector) are equivalent to radiance errors. The CHRONOS implementation provides nearly simultaneous acquisition of the gas and vacuum cell signals through a common optical path, and minimizes ground co-registration errors between signal pairs. Observation simulation studies using representative GEO spacecraft pointing data have been performed to determine the effect of 'jitter' in spacecraft pointing during the acquisition of a signal pair. The displacement between a single paired gas/vacuum measurement is limited to ≤5 μrad to ensure acceptable changes in ground pixel reflectance based on MOPITT experience (Deeter et al., 2011). This requirement corresponds with a gas cell-to-vacuum cell frame time limited to 60 msec, readily achievable with a physically realistic cell size and rotation frequency, frame acquisition and readout rate. The large (>3000 kg) size of a commercial communications spacecraft therefore serves to naturally attenuate jitter sources over very short time frames, avoiding the need for a costly image stabilization subsystem.

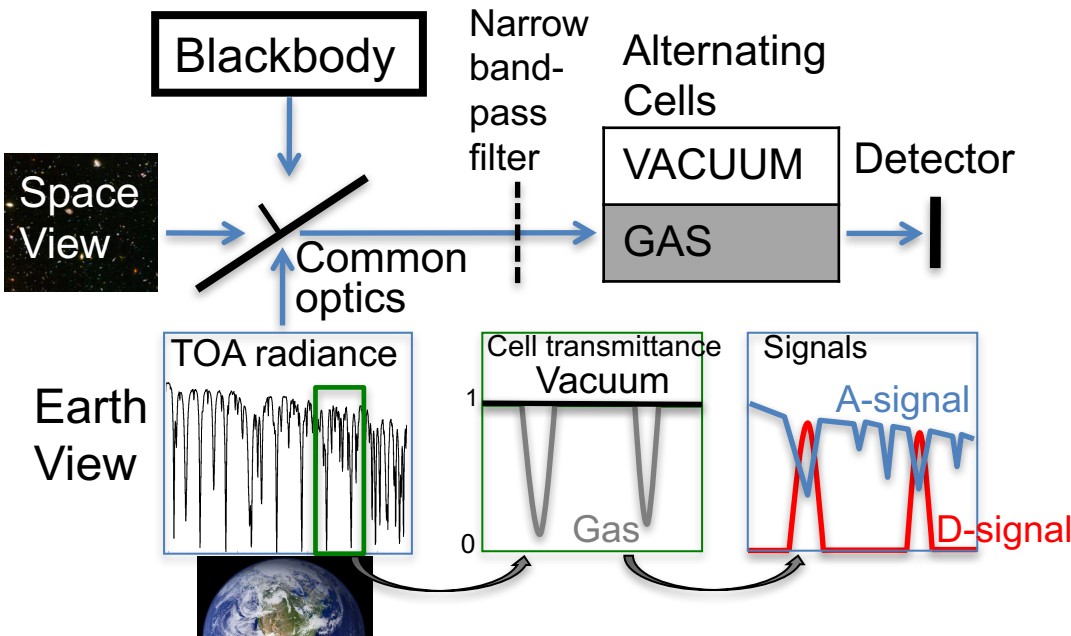


**Figure 4:** Simplified depiction of the CHRONOS GFCR measurements to show how average (A) and difference (D) signals are generated from spectrally-correlated, band-integrated radiance measurements through the vacuum (V) and gas (G) cells. Upwelling atmospheric radiance passes through a narrow bandpass filter, selected for the target gas spectral range, a target gas cell, and

on to a detector pixel. For CHRONOS, within 60 msec, the atmospheric radiance passes through an identical bandpass filter, an identical reference vacuum cell, and falls on the same detector pixel.

In gas filter correlation radiometry, the relationship of the instrument analog signal and the actual spectrum must be interpreted using a forward atmospheric model and line-by-line spectral

radiative transfer calculations (Pan et al., 1995). For instrument development, these calculations form the basis of the instrument spectral characteristics definition (bandpass and width for each target gas and spectral region), and quantify the instrument sensitivity to the target gas, the effects of signatures of non-target gases in the selected spectral region, and the effects of variations in the underlying surface temperature, emission, and reflectivity. After launch, these calculations are a

crucial part of the instrument model used in data retrieval.

## 3.2 Spectroscopy of CO and CH$_4$ and the CHRONOS Instrument Signals

Two CO spectral bands, the mid-wave infrared (MWIR) fundamental at 4.6 µm (Figure 5) and the short-wave infrared (SWIR) overtone band at 2.3 µm (Figure 6), are the only spectral regions that produce CO features easily distinguished from the surrounding spectra at wavelengths shorter than

microwave, and thus are useful for passive remote sensing of tropospheric CO (e.g., Edwards et al., 1999; 2009). Measurements in the MWIR band rely on thermal emission from the Earth's surface and atmosphere (that can be obtained both day and night), and relatively strong spectral features. Measurements in the MWIR are only sensitive to changes in lower atmosphere CO concentration when sufficient thermal contrast exists between the surface and near-surface

atmosphere (Deeter et al., 2004). Typically, MWIR signals are most sensitive to CO concentration changes in the mid-troposphere, where long-range pollution transport typically occurs. In contrast, measurements in the CO SWIR band rely on solar radiation reflected from the Earth's surface in daylight, with comparatively weak CO spectral features (Deeter et al., 2009). Typically, the SWIR signal has almost uniform sensitivity to changes in the CO vertical profile, including information

near the surface.

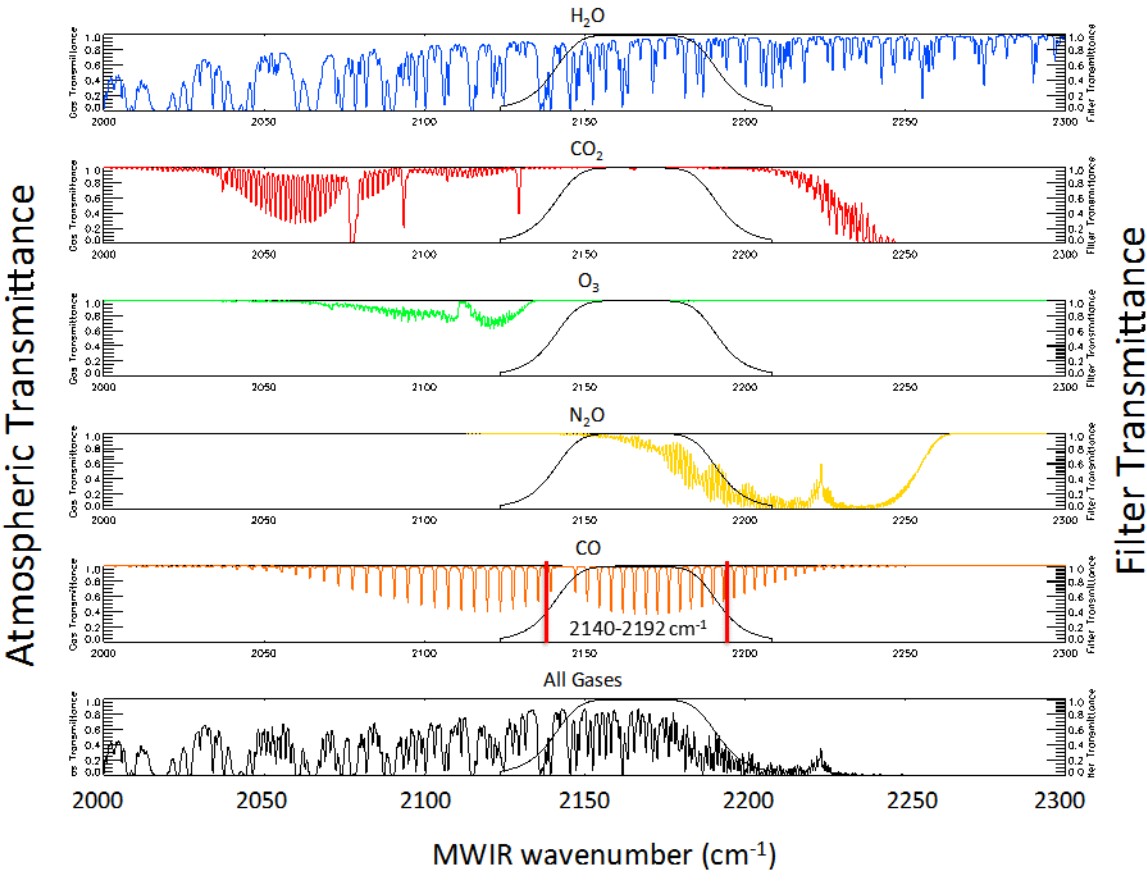

**Figure 5:** Atmospheric transmittance for primary trace gases in the MWIR vs. wavenumber. The CHRONOS filter transmission is indicated by smooth bandpass curves with solid red lines at filter half-power points (50% transmittance). CHRONOS measures only CO in the MWIR.


Several spectral bands may be considered for retrieving $CH_4$. Infrared measurements near 7.7 μm (e.g., Payne et al., 2009) generally lack sensitivity to near-surface $CH_4$, similar to MWIR CO. Both SCIAMACHY (e.g., Frankenberg et al., 2005; 2011, Wecht et al., 2014b) and GOSAT (e.g., Morino et al., 2011, Schepers et al., 2012) have produced $CH_4$ data products using reflected

sunlight in the SWIR to obtain a true total column. CHRONOS will also measure a 2.2 μm SWIR $CH_4$ band, shown in Figure 6.

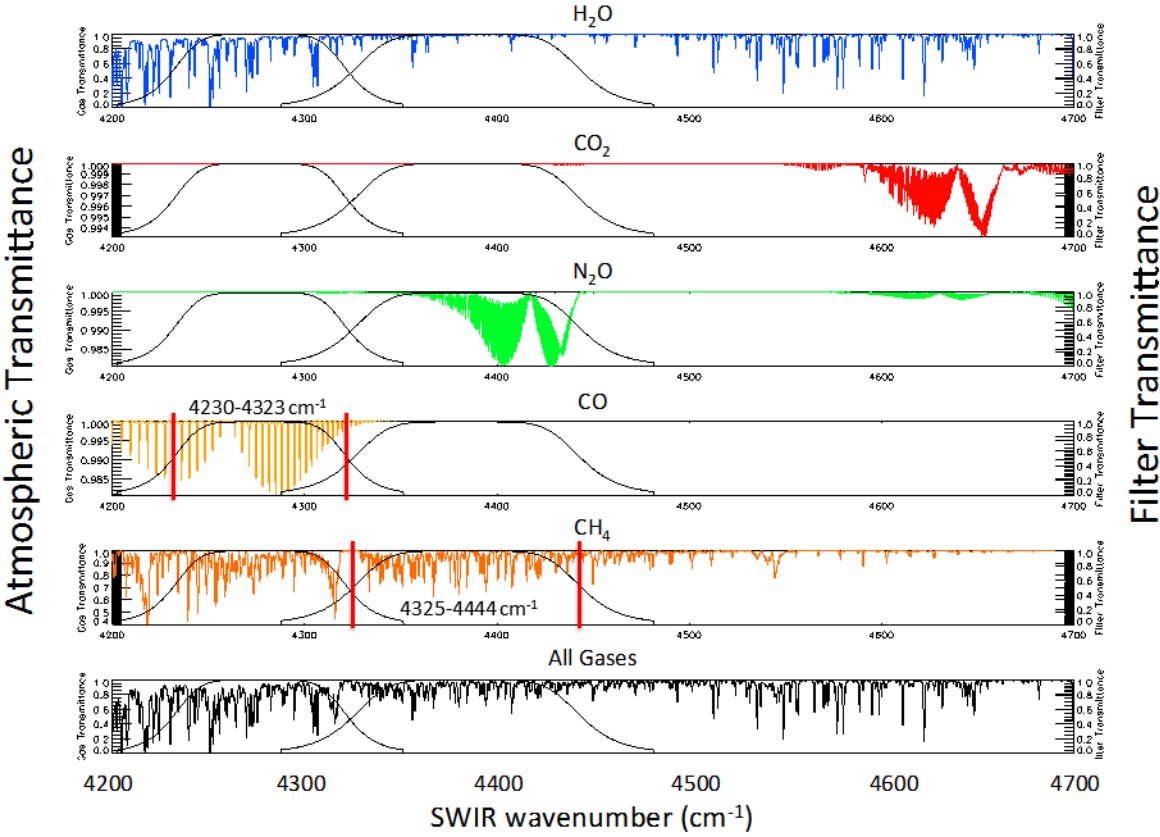

**Figure 6:** Atmospheric transmittance for primary trace gases in the SWIR vs. wavenumber. CHRONOS filter transmission is indicated by smooth bandpass curves with solid red lines at filter half-power points (50% transmittance). CHRONOS measures both CO and $CH_4$ in the SWIR.

The GFCR instrument generates spectrally-correlated, band-integrated radiance measurements through alternate gas and vacuum cells, producing radiance pairs. As shown in Figure 4, the Difference (D-signal), constructed by differencing the gas and vacuum cell radiances, contains spectral contributions only from the target gas absorption line positions within the spectral passband. The Average (A-signal), the mean of the gas and vacuum cell radiances, has a spectral contribution that is low at the target gas line positions and high elsewhere. As such, the A-signal carries background information on the FOV scene characteristics. Therefore, the ratio of the D-signal and A-signal, D/A, eliminates the background radiance term and reduces the impact of uncertainties associated with surface reflectance, interfering gases, or optically thin aerosols and clouds. In non-optically thin cases of clouds or aerosols (OD ≳ 0.2, identified using the GOES ABI cloud mask for example), data are discarded, and no retrieval is performed. This approach is

possible due to the high temporal and spatial sampling of CHRONOS and the availability of ancillary cloud and aerosol geostationary observations, both current and expected (e.g., Heidinger, 2011).

The designated pressure in the instrument cell determines the width of the fill-gas spectral lines, and thus the effective spectral sampling resolution of the correlation filter. We note that the CHRONOS MWIR CO and SWIR $CH_4$ bands also contain water vapor ($H_2O$) and nitrous oxide ($N_2O$) absorption features (Figures 5 and 6). The gas correlation removes the absorption effects of these interfering gases for the D-signal. The interference of $H_2O$ and $N_2O$ in both the MWIR and SWIR channel A-signals is modeled in the forward model radiative transfer algorithm using analyzed $H_2O$ concentration fields from meteorological data and inferred $N_2O$. $N_2O$ is a long-lived gas (~120 years) with predictable variability (Angelbratt et al., 2011).

SCIAMACHY and GOSAT $CH_4$ SWIR retrievals are sensitive to scattering by dust, aerosols and thin cirrus (Gloudemans et al., 2008; Schepers et al., 2012) and address these errors by using $CO_2$ (with known abundance) as a proxy for the scattering effects or by performing a physical retrieval of effective parameters for the scattering layer. For GOSAT $CH_4$ data, these two approaches yield similar precision (~17 ppb) and biases less than 1% compared to TCCON (Wunch et al., 2010), but with lower bias for the proxy method (Schepers et al., 2012). In the proxy retrieval using $CO_2$, the dry mole fraction of $CH_4$ ($x_{CH4}$) is computed by $x_{CH4} = \frac{[CH4]}{[CO2]} x_{CO2}$ where $[CH4]$ and $[CO2]$ are the retrieved columns from spectral radiances that are close in wavenumber and $x_{CO2}$ is the dry mole fraction computed from a global model of atmospheric $CO_2$ (Frankenberg et al., 2005; Schepers et al., 2012). This method assumes that aerosol scattering modifies the light path for $CO_2$ and $CH_4$ spectral absorption in the same way, and that model values for $x_{CO2}$ are accurate.

Retrievals with GFCR measurements are similar to the "proxy retrieval" but they correct the input radiance instead of the retrieved column, and do not make assumptions about aerosol scattering in different spectral bands or rely on knowing $CO_2$ abundance. CHRONOS uses the D/A signal ratio where D and A are both modified in the same way by aerosol scattering, which has a smooth spectral behavior over the CHRONOS bandpass. For optically thin aerosol and cloud scenes, this ratio gives an accurate total column amount, but to compute a dry mole fraction (xCH4), we require additional information about the surface pressure (for example, from GOES-16 meteorological data) in order to estimate the dry air column. In general, GFCR retrievals are more resilient than

spectral radiance measurements to errors in surface and contaminant species assumptions due to the use of radiance differences and ratios (Pan et al., 1995).

**3.3 Measurement Radiometric Accuracy and Precision**

By using D/A signal ratios, the GFCR technique is inherently less sensitive to radiance bias errors than spectrometer measurements. However, three primary sources of potential retrieval bias remain: surface albedo spectral variability, aerosol scattering, and water vapor errors in meteorological data, which are typically < 10% for N. America (Vey et al., 2010). Spectral

variation in surface albedo proved to be a significant obstacle for MOPITT $CH_4$ retrievals (Pfister et al., 2005). This was because of the width and spectral location of the MOPITT passband, combined with changing scene albedo arising from LEO spacecraft motion during the acquisition of a single measurement (Deeter et al., 2011). For CHRONOS, the $CH_4$ passband has been optimized in both width and spectral location (Table 1) to mitigate these errors.

For a GFCR, the radiance precision needed to measure a change in column is given by $\Delta D/A$, for D and A defined above, where $\Delta D$ is determined using the instrument sensitivity to the column change ($\partial D/\partial col$) (Pan et al., 1995). Profile or column retrieval precision requirements are achieved in ground processing by averaging geo-located, cloud screened radiances for three minutes (375 separate gas-vacuum measurements for each product: CO [4.6 μm, 800 hPa], CO

[4.6 μm, 200 hPa], CO [2.3 μm, 100 hPa]; and 750 measurements of $CH_4$ [2.2 μm, 800 hPa]). A single retrieval for each product is performed on these averaged radiances. The process of averaging radiances and then retrieving products is repeated for all data acquired in the 9.7-minute data acquisition period. Table 1 lists the modeled signal-to-noise (SNR) and the total number of individual data acquisitions in each pixel in the 2D detector array ("frames") obtained in a single

9.7-minute data acquisition period, for the minimum radiance case defined from MOPITT on-orbit radiance records. This minimum SNR provides at least 30% margin for meeting the radiance precision requirements.

**Table 1.** The multi-layer dielectric optical coatings on the CHRONOS gas cell windows define

the center wavelength and bandpass. Each spectral coating and cell pressure is identified through modeling to provide the optimal measurement. The signal-to-noise ratio (SNR) listed provides at

least 30% margin over the SNR required to achieve radiance precision. All data frames are obtained within a single 9.7-minute data acquisition period.

| Cell | Gas Filter | Center λ (μm) | Cell Pressure (hPa) | Band Pass (μm) | Band Pass (cm$^{-1}$) | Co-added SNR at minimum radiance | Number of frames obtained |
|------|-----------|---------------|---------------------|----------------|-----------------------|----------------------------------|---------------------------|
| 1 | CO | 4.6 | 200 | 4.562 – 4.673 | 2140 – 2192 | 595 | 95 |
| 2 | Vacuum | 4.6 | 0 | 4.562 – 4.673 | 2140 – 2192 | -- | -- |
| 3 | CO | 4.6 | 800 | 4.562 – 4.673 | 2140 – 2192 | 595 | 95 |
| 4 | CO | 2.3 | 100 | 2.313 – 2.364 | 4230 – 4323 | 3255 | 889 |
| 5 | Vacuum | 2.3 | 0 | 2.313 – 2.364 | 4230 – 4323 | -- | -- |
| 6 | CH$_4$ | 2.2 | 800 | 2.250 – 2.312 | 4325 – 4444 | 4390 | 1012 |
| 7 | Vacuum | 2.2 | 0 | 2.250 – 2.312 | 4325 – 4444 | -- | -- |
| 8 | CH$_4$ | 2.2 | 800 | 2.250 – 2.312 | 4325 – 4444 | 4390 | 1012 |

CO profile retrievals require 10% precision to capture the fine-scale space and time variability of CO and quantify transient sources (Fishman et al., 2012; Emmons et al., 2009). Based on GEO-CAPE CH$_4$ emission OSSEs (Wecht et al., 2014a), monthly emissions estimates with <10% error on county-level spatial scales (~40 km x 40 km) require a daily precision on averaged retrievals of total column CH$_4$ <1%. CHRONOS will deploy two identical CH$_4$ channels with combined 0.7%
precision for a 9.7-minute data acquisition that exceeds the GEO-CAPE daily requirement. The CHRONOS sub-hourly CH$_4$ sampling capability and the relatively slow rate of change in CH$_4$ column abundance enable the combining of samples to further improve CH$_4$ column precision, allowing identification of CH$_4$ changes on daily scales, and verification of state and federal pollution reduction goals (Miller et al., 2013). As discussed in Section 3.2, a major advantage of
the GFCR measurement technique is the ability to eliminate any contaminating signal that is not spectrally correlated with the lines of the cell target gas. In the spectral regions utilized by CHRONOS, water vapor spectral lines are ubiquitous, and in the SWIR, the effects of aerosol must be considered. Figure 7 shows CHRONOS simulated A and D signals for mid-latitude summer atmospheric conditions (Anderson et al., 1986), with and without aerosol scattering. The

VLIDORT radiative transfer model (Spurr, 2006) is used for modeling the aerosol scattering, and the OPAC (Optical Properties of Aerosols and Clouds) database (Hess et al., 1998) provides AOD adjusted to 2.25 µm. The case shown in Figure 7 is for AOD that is 50% larger than the OPAC urban aerosol case. Based on the simulated retrievals we perform, 1% errors in total column correspond to 0.2% errors in D/A. The nominal urban aerosol loading considered in OPAC would
lead to ~ 0.026% errors in D/A, which translates to a ~0.13% error in total column. Similar errors in D/A due to aerosol scattering are obtained for the CHRONOS 2.3 µm CO channel, and can then be compared to MOPITT measurement errors in D/A that are around 1 to 2% for scenes with minimal geophysical noise. The insensitivity of D/A to aerosol scattering is found to hold for a large range of aerosol types and loading, with the largest errors (up to 0.3%) due to desert dust,
consistent with Gloudemans et al. (2008). Expected errors due to uncertainties in water vapor were also simulated using perturbations of the mid-latitude summer atmosphere and are < 1% for CO and < 0.2% for $CH_4$.

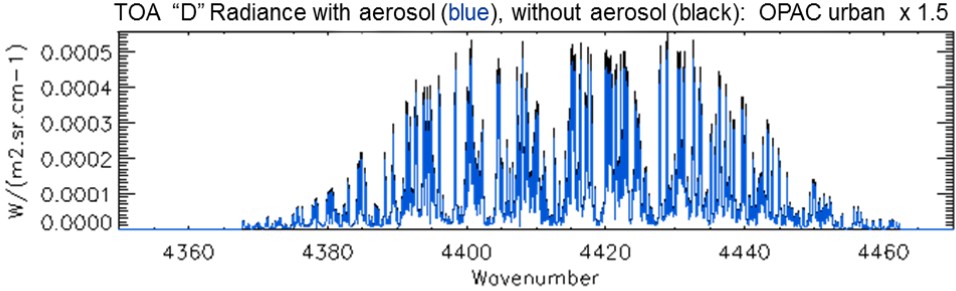

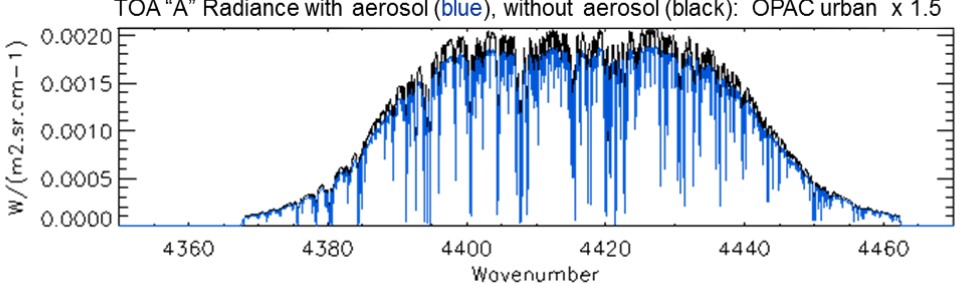

**Figure 7:** Forward model results with aerosol loading. Simulated radiance spectra for CHRONOS
corresponding to TOA D (top panel) and A (bottom panel) signals with the CHRONOS $CH_4$ SWIR channel bandpass applied. Simulations are for a mid-latitude summer atmosphere with solar zenith angle = 0, satellite zenith angle = 40° and surface albedo = 0.2. Black lines represent the case without aerosol scattering and blue lines show radiances with aerosol scattering for urban aerosols

(AOD is 0.089, which is obtained by scaling the OPAC urban aerosol case by 1.5). D/A is
computed after integration over the bandpass and is changed by -0.039% for the case with aerosols
compared to without.

A summary of CHRONOS precision and accuracy requirements for column CO and $CH_4$ is given
in Table 2. Validation activities for CHRONOS will use aircraft profiles from on-going flight
programs, such as IAGOS (Nédélec et al., 2015) and existing ground data networks such as
TCCON (Wunch et al., 2010) to detect biases in CO and $CH_4$ similar to the proven approach used
for GOSAT and OCO-2 validation (Schepers et al., 2012).

**Table 2. Expected precision and accuracy for CHRONOS.**

| Column Error Source | MWIR CO (night) | MWIR+SWIR CO (day) | SWIR $CH_4$ (day) |
|---|---|---|---|
| Precision requirement<br>MOPITT performance<br>Corresponding SNR (A/ΔD) | <10%<br>5-15%<br>457 | <10%<br>2-10%<br>2499 | <0.7%<br>n/a<br>3374 |
| Radiometric bias | <0.1% | <0.1% | <0.1% |
| 10% water vapor error | <0.7% | <0.7% | <0.15% |
| Albedo variation | Negligible for MWIR CO band[1] | <0.06% | <0.1% |
| Urban aerosol loading | Negligible in MWIR[2] | <0.03% | <0.13% |

[1](MWIR CO band is 0.11 μm wide; based on MOPITT experience, no significant errors due to albedo spectral
variation); [2](e.g., Russell et al., 1999, Bohren and Huffman, 1983)

## 4 The CHRONOS Instrument and Operation

The CHRONOS measurement domain, shown in Figure 8, extends over North America and
includes adjacent oceans in order to observe pollution inflow and outflow using observations in
the MWIR CO channels. In the SWIR channels, sunlight is mostly absorbed in the ocean, and no
trace gas retrievals are expected over the ocean in the SWIR channels. The CHRONOS ground
sample area varies gradually across the field of view due to curvature of the Earth as seen from
GEO, with smaller than 4 km x 4 km (16 $km^2$) nominal pixel area at the center of the domain,
increasing to 19.3 $km^2$ at the edge of the CONUS domain (e.g., Seattle). This spatial resolution
enables emissions estimates at the U.S. county scale even for the smallest county in the continental

U.S., New York County (i.e., Manhattan), NY, which contains 3.5 CHRONOS pixels. The increase in pixel size toward northern latitudes is commensurate with the increasing scale of dominant emissions sources, such as large-scale wetlands in Canada (e.g., Pickett-Heaps et al., 2011). To account for these variations, CHRONOS Level 2 (individual retrieval) data will be re-gridded (Vijayaraghavan et al., 2008; Guizar-Sicairos et al., 2008) in Level 3 (gridded) data to facilitate user scientific analysis using standard tools.

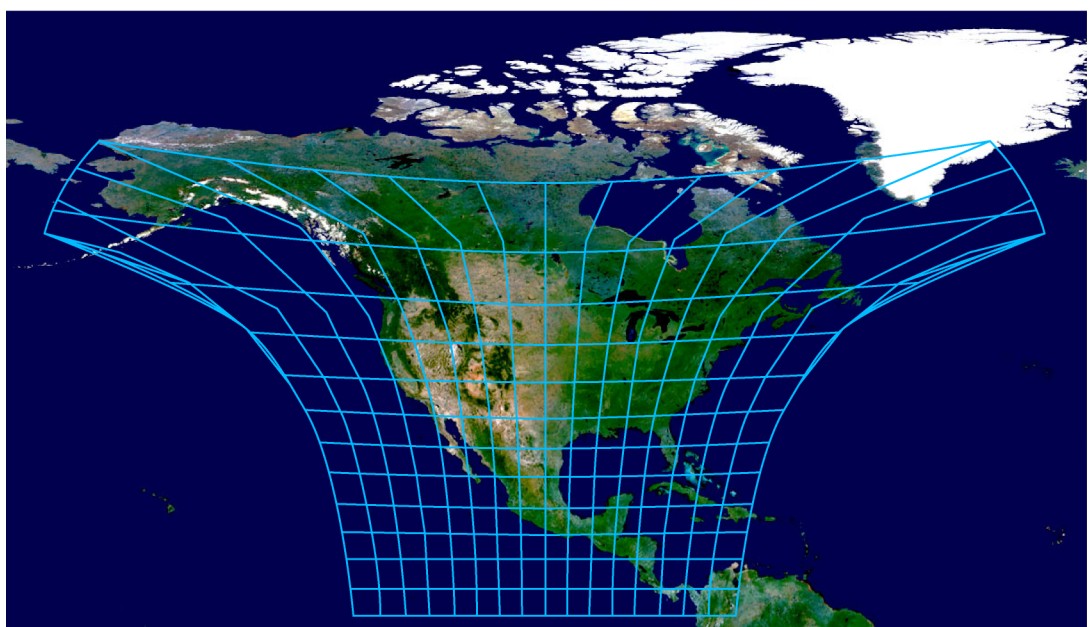

**Figure 8:** CHRONOS field of view from geostationary orbit at 100° W. Each grid cell above represents 125 x 125 pixels. All pixels are acquired with full precision within the ~10 minute CHRONOS data acquisition.

The CHRONOS GFCR is a staring infrared 2-D camera with a continuously rotating wheel that houses gas and vacuum cells, which sequentially pass through the optical path. Figure 9 depicts the instrument, which comprises optomechanical, calibration, focal plane, thermal, and control electronics subsystems. Within the optomechanical subsystem, the gas cell filter wheel assembly contains cells as specified in Table 1. A selection mirror determines the source of the input radiance being filtered by the cells and imaged by the optics (Earth view, on-board calibration subsystem, a deep space view, and a blocked or closed position).

CHRONOS uses an all-digital (Brown et al., 2010) cryogenically cooled HgCdTe large area focal plane array to detect the spectral radiance. The CHRONOS instrument has been designed around

commercially available, space proven, radiation-hardened large format focal plane arrays (e.g., flown on India's Chandrayaan-1 mission/NASA Moon Minerology Mapper (Green et al., 2011), DOD's CHIRP experiment (Levi et al., 2011), and NASA's Near Infrared Camera on the James Webb Space Telescope (Garnett et al., 2004)). Low dark current ($6.2 \times 10^9$ e-/cm$^2$-s at 110 K), low

readout noise (high gain: 40 e- rms; low gain: 200 e- rms), high, stable quantum efficiency (0.7 at 2.2, 2.3, and 4.6 µm) and fast electronics are necessary characteristics for this application. For a 2-D imager such as CHRONOS, the pixel format (presently 2048 x 2048) and the desired observational domain determine the single pixel ground sample area from geostationary orbit.

A small, high-reliability, space-proven cryocooler cools the focal plane array and a portion of the

optics module. Instrument control electronics provide the functionality to receive communications (commands) from the host spacecraft, control the instrument, sequence the data collection operations, and ultimately send science data to the host for downlink.

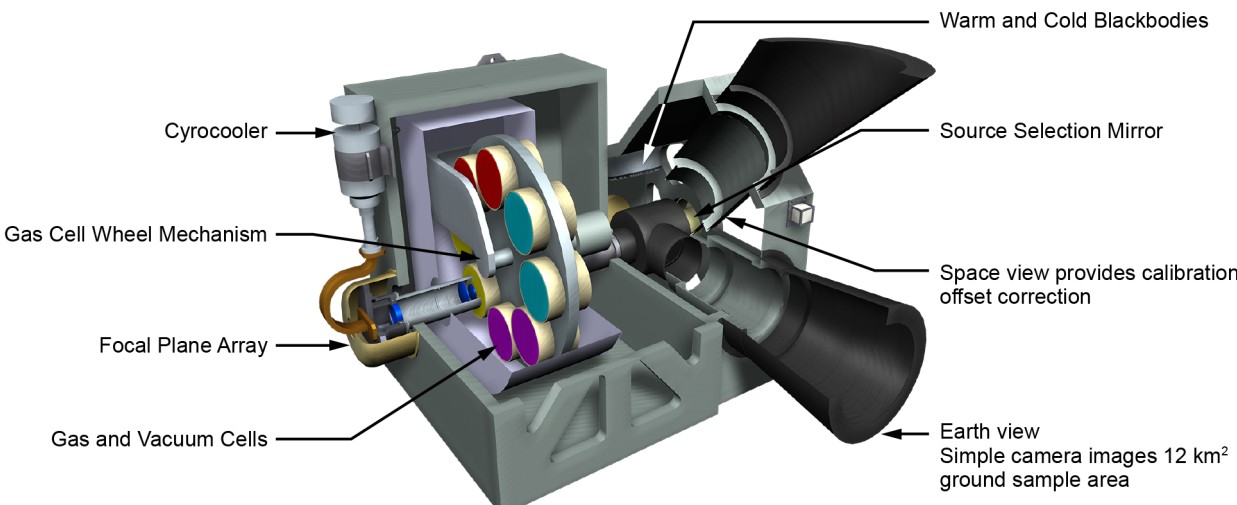

**Figure 9:** The CHRONOS GFCR is a staring infrared camera with gas cell filters in the optical

path. A source selection mirror determines the input to the system (Earth FOV, one of three onboard calibration sources, deep space, or closed). Optics image this source input onto a cryogenically cooled large area focal plane array.

Figure 10 shows the image collection timing between a gas cell and its physically adjacent paired

vacuum cell on a continuously rotating wheel. When an unobscured FOV emerges as a cell rotates through the optical path, the focal plane collects an image of the entire physical domain using one of two integration times (corresponding to low gain and high gain). Multiple gains are necessary

to image the high dynamic range across the entire FOV with the required signal-to-noise ratio (SNR). Only 60 msec later, the FOV of the next cell, (a vacuum cell in the case of Figure 10), is unobscured and the focal plane collects an image. The short 60 msec between images effectively freezes the scene, allowing the GFCR algorithm to process the pair cell and vacuum signals together without geometric corrections, and providing nearly simultaneous gas and vacuum cell views described in Section 3.1. Single frames of paired gas and vacuum cell signals, as described above, are continuously collected until a prescribed number of images have been collected for each gas/vacuum cell pair. All of the images are downlinked through the host spacecraft. In ground processing, the single frame Level 0 (signal count) data are processed for image registration and radiance calibration before being co-added to build up the required signal to noise ratio for the Level 1 (radiance) measurement at each location (pixel). The full data collection sequence includes calibration views, the full Earth view image collection outlined above for both low and high gains, followed again by calibration. The required SNRs for all channels are achieved in 9.7 minutes of measurement time. The CHRONOS gas cell filter wheel rotates continuously, and data may be obtained continuously, for up to 6 full-precision data takes per hour. Parameter tables can be uploaded to alter this sequence, or command additional data collects as necessary.

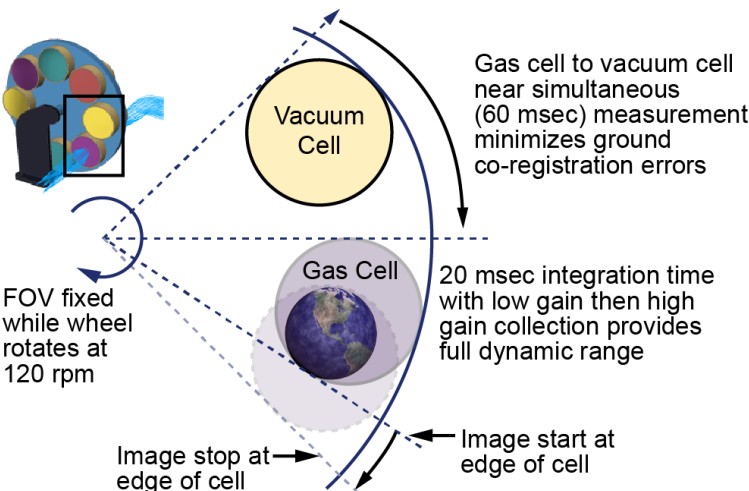

**Figure 10:** CHRONOS' eight gas cells are mounted in a continuously spinning mechanism wherein each cell in sequence exposes an unobscured Earth FOV as defined in Figure 8. A single frame image is collected with a prescribed integration time. Single frames are continuously collected and downlinked via the host spacecraft. In ground processing the ensemble of single frames are co-added to achieve the required signal to noise ratios for each measurement.

For on-orbit radiance calibration, CHRONOS views high-precision hot and cold black bodies and deep space for the MWIR channels, and a tungsten lamp (LandSat Operational Land Imager heritage) and a closed aperture for the SWIR calibration within each 10-minute data acquisition.

CHRONOS cloud detection will follow the MOPITT algorithm approach, which uses the MWIR A-Signals as the primary test for the presence of cloud, based on observed brightness temperature
(Warner et al., 2001). In the case of MOPITT operation, cloud flags are then verified with the Moderate Resolution Imaging Spectroradiometer (Terra/MODIS) cloud data products, when available. Using a similar approach, CHRONOS will use the GOES-R Advanced Baseline Imager (ABI) cloud mask (Heidinger, 2011) to verify cloud detection. Cloud movement is assumed negligible during a 60 msec frame measurement. MOPITT retrieval experience shows that the
GFCR technique can tolerate up to ~5% cloud contamination and still treat the pixel as cloud-free (Warner et al., 2001). While the approach of using D/A for retrievals discussed in Section 3.3 will cancel some of the errors due to undetected aerosols or clouds (e.g., thin cirrus), remaining retrievals errors (e.g., O'Dell et al., 2011), particularly for $CH_4$, will require further study using both CHRONOS radiances and GOES-16 ABI observations. Combined with CHRONOS' sub-
hourly revisit, the small nominal ground sample area increases the probability of obtaining cloud-free pixels in regions of broken cloud. This is an advantage compared to observations from LEO where a cloud-free scene may not be encountered at a given location over several days. Figure 11 shows OSSE results for simulated CHRONOS observations over a 2-week summer period. This study indicates that 70–75% of 4 km x 4 km pixels can be treated as cloud-free in the West and
Central U.S. and 60–65% in the East U.S.

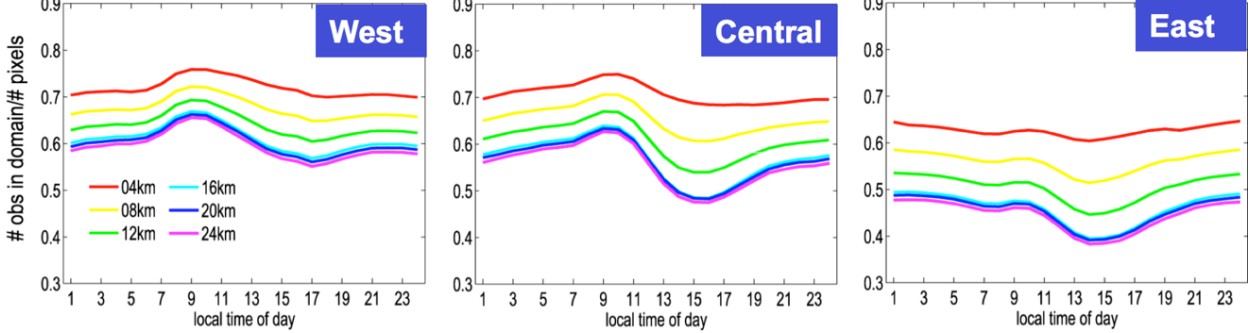

**Figure 11:** CONUS cloud statistics from OSSE results for 15-30 July 2006 using a high spatial resolution WRF-Chem run and a GFCR instrument with allowable pixel cloud fraction set at 3%. For different geographical regions, the fraction of cloud-free scenes (the number of cloud-free

pixels observed as a fraction of the total number of pixels in the region) is plotted for different assumed pixel sizes; red represents the CHRONOS 4 km x 4 km pixels. Clouds are defined by 4-km grid integrated total hydrometeors $> 10^{-8}$ kg/kg.

After cloud detection, the retrieval algorithm accesses current best meteorological analysis data
for surface pressure and temperature, atmospheric temperature and water vapor profiles to include in the forward model radiance calculation. A maximum a posteriori retrieval (Rodgers, 2000) is then used to convert Level 1 TOA radiances to Level 2 vertical trace gas distributions.

## 5 Characterization of CHRONOS CO and CH$_4$ Retrievals

### 5.1 Multi-spectral CO Measurements and Vertical Profile Information

CHRONOS CO measurements use MWIR thermal emission (day and night), with sensitivity to free tropospheric CO, and SWIR solar reflection (day), with sensitivity to total column CO. These measurements are combined in a multispectral retrieval to obtain vertical profiles of CO abundance, Figure 12. Following MOPITT retrieval algorithms, CHRONOS will employ the maximum a posteriori methods of Rodgers (2000), which provide an averaging kernel (AK) that
represents the sensitivity of the retrieval to the abundance of the target trace gas in each retrieval pressure layer in $\log_{10}$ of volume mixing ratio (Deeter et al., 2007). The single pixel retrieval results depend on both the choice of a priori profile and a priori error covariance, and retrieval diagnostics such as the averaging kernel and the posterior error covariance depend on the a priori error covariance. MOPITT retrieval algorithms, since version 4, have applied spatially and
monthly (but not yearly) varying a priori profiles from a model climatology, and a single prior error covariance with diagonal values corresponding to 30% variability in fractional volume mixing ratio and a correlation height (off-diagonal variation) of 100 hPa (Deeter et al., 2010). CHRONOS retrievals will emulate the MOPITT retrieval approach to facilitate comparisons and analyses of long-term changes in CO.

Degrees of freedom for signal (DFS) in the retrieval are computed from the trace of the AK, and provide a measure of the independent profile information available. DFS values are ~1.5 to 2 for retrievals using only MWIR channels with CO gas cells at two different pressures, while a column retrieval with the SWIR channel alone has at most a DFS of 1. The CHRONOS multispectral

retrievals have DFS values typically > 2. Figure 13 shows CO retrieval results from MOPITT that compare the sensitivity of MWIR-only, SWIR-only and multispectral retrievals. We note that MOPITT retrievals would have higher values of DFS without the presence of geophysical noise in MOPITT observations (Deeter et al., 2011). Geophysical noise is introduced by changes in the FOV surface albedo due to LEO spacecraft motion during the time taken for MOPITT signal acquisition. The CHRONOS stationary FOV and single frame integration time of 20 msec mitigates this source of noise. Multispectral CO retrievals from MOPITT have demonstrated the improvements in sensitivity to surface layer CO abundance (Worden et al., 2010), have been validated (Deeter et al., 2011; 2013), and used in many studies to distinguish surface pollution emissions from transported plumes (e.g., Worden et al., 2012; Jiang et al., 2013; 2015; He et al., 2013; Silva et al., 2013; Worden et al., 2013; Huang et al., 2013; Anderson et al., 2014; Bloom et al., 2015). The CHRONOS multispectral retrievals would extend the MOPITT record of vertical layers of CO over North America when MOPITT is finally decommissioned, since MOPITT is the only satellite mission to demonstrate multispectral trace gas retrievals from a single space-based instrument. The multispectral retrieval approach for CO allows for up to 3 DFS, which is a practical upper limit on CO vertical information based on atmospheric radiative transfer. As has been demonstrated by other on-orbit sensors measuring CO, an instrument design with more gas cells, or a spectrometer with arbitrarily fine spectral resolution (George et al., 2009), does not produce retrievals with greater DFS. Thus, CHRONOS would produce the maximum vertical information possible for CO with a passive sensor.

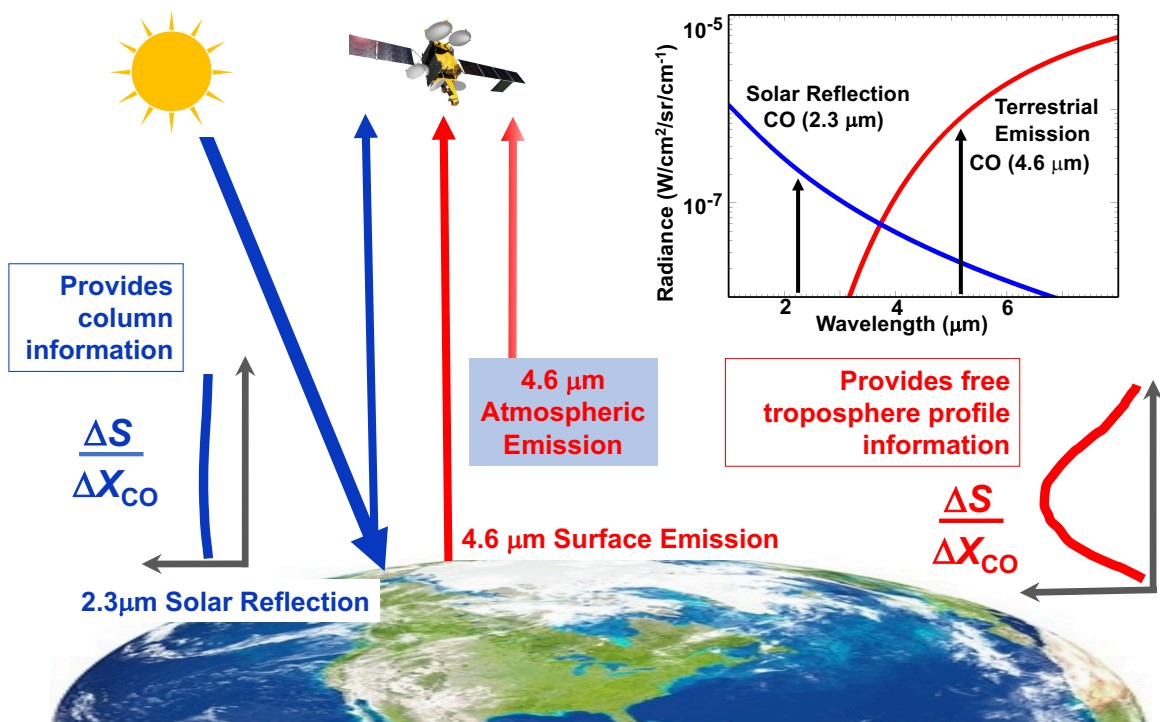

**Figure 12:** Physics of CHRONOS and MOPITT multispectral measurements. In the SWIR at 2.2 and 2.3 μm, measurement signals rely on daytime reflected solar radiation and weak spectral features. Changes in $CH_4$ and CO mixing ratios, $\Delta X$, produce uniform signal sensitivity, $\Delta S$, throughout the vertical profile, including near the surface. In the MWIR at 4.6 μm, signal sensitivity is greatest in the middle troposphere, except in cases of high thermal contrast between the surface and the lowest atmospheric layers. CHRONOS $CH_4$ SWIR retrievals use the solar reflected radiance to provide a true total column, and CO multispectral retrievals combine the SWIR and MWIR measurements to increase the sensitivity to near-surface CO. While this increased sensitivity varies depending on scene characteristics such as albedo, in many cases, it provides improved information to distinguish local air pollution emissions and transported plumes.

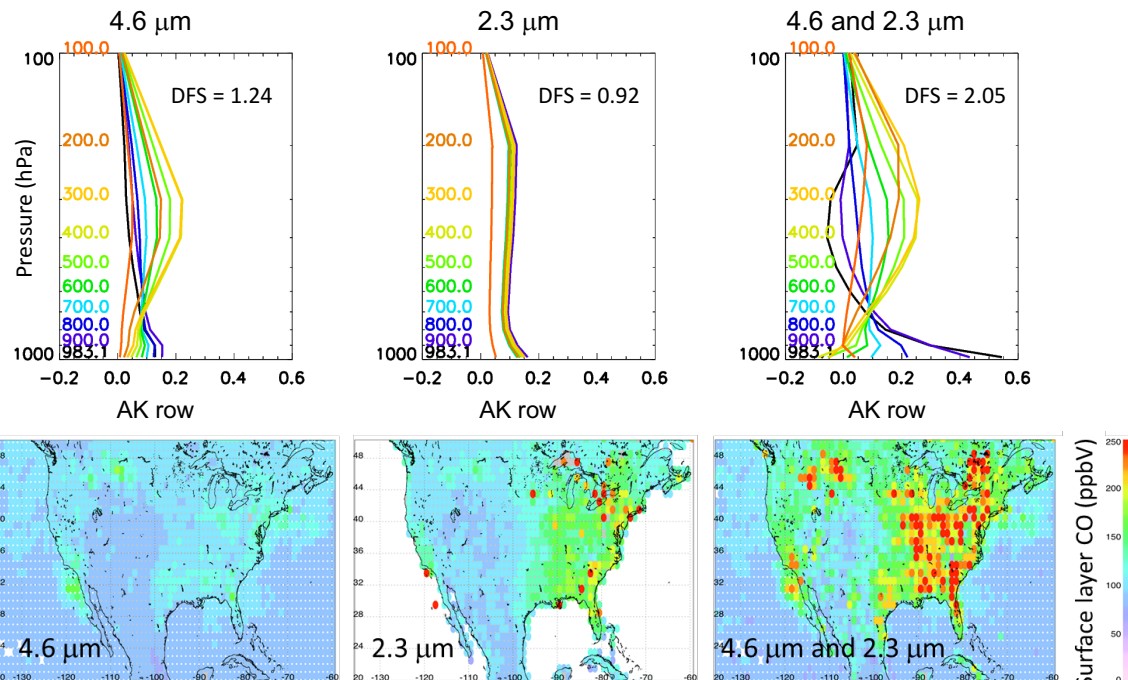


**Figure 13:** Comparison of surface layer CO for MOPITT version 7 data (Deeter et al., 2017) MWIR (4.6μm), SWIR (2.3 μm) and multispectral (MWIR and SWIR combined) retrievals for Aug. 2000. Top panels show representative averaging kernel (AK) rows, where line colors indicate the pressure layers given on the left side of the panels, for the three retrieval types (location at

30.72°N, 96.50°W). Bottom panels show maps of the surface layer CO abundance indicating how detailed information is obtained in the multispectral retrievals, but is absent in the single channel retrievals.

**5.2 Retrieval Sensitivity to Near-Surface CH$_4$**

The vertical profiles of CH$_4$ are similarly characterized using the AK from maximum a posteriori retrievals (Rodgers, 2000). Radiative transfer modeling has been developed to compute weighting functions, i.e., radiance Jacobians integrated over the filter bandpass to assess the sensitivity to changes in the CH$_4$ column. Figure 14 shows an example of a simulated CHRONOS CH$_4$ weighting function and corresponding AK (see caption for simulation assumptions). For the SWIR

measurements, the signal source is solar reflectance with a measurement sensitivity response that is nearly uniform in the vertical, giving true total column CH$_4$ information with DFS close to 1.

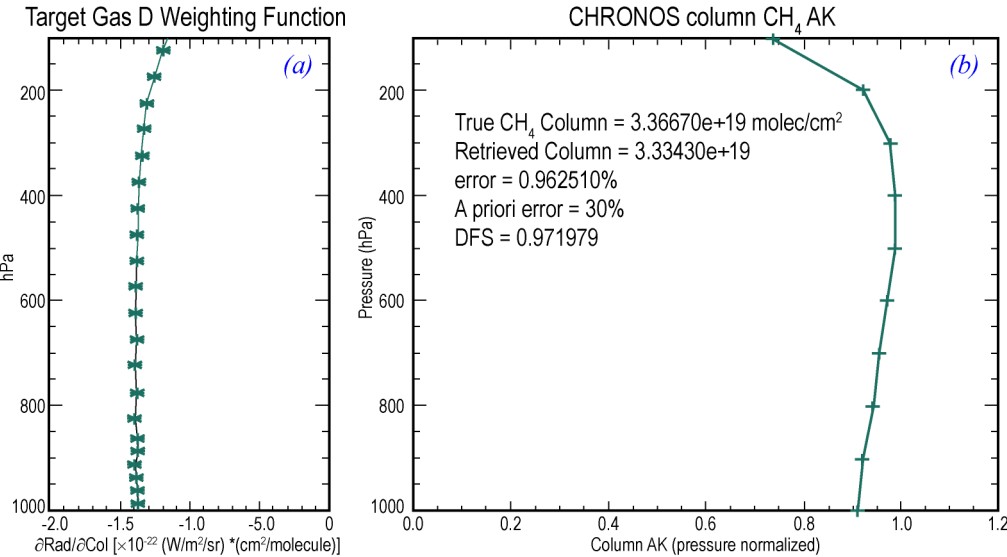

**Figure 14:** CHRONOS $CH_4$ D-signal weighting function and retrieval column AK. (a) Sensitivity of the D-signal to changes in $CH_4$ column ($\partial\mathbf{Rad(D)}/\partial\mathbf{col}$) as a function of vertical pressure for a standard mid-latitude summer atmosphere with albedo = 0.1, SZA = 0°, satellite ZA = 40°. (b) The retrieval column averaging kernel from the corresponding D/A signal ratio and Jacobian. This assumes CHRONOS measurement precision and a priori covariance with 30% diagonal errors and 500 hPa correlation length, (retrieval was performed on a coarser pressure grid than the weighting function calculations). Since the signal source is solar reflectance, the response is nearly uniform vertically with DFS close to 1.

The magnitude and shape of the column $CH_4$ AKs have only small variations with input atmospheric parameters (such as temperature and water vapor) and input surface parameters such as albedo (assuming non-zero albedo). However, there is a more significant dependence of the $CH_4$ AK on parameters that depend on the total amount of $CH_4$ along the observation path, such as solar zenith angle (SZA), satellite zenith angle and $CH_4$ abundance. Figure 15 shows how the sensitivity to $CH_4$ in the lowest (near-surface) layer depends on SZA for A, D and D/A signals. While the A and D signals both have reduced sensitivity with higher SZA, as expected for solar reflection radiances, the D/A ratio sensitivity increases slightly, with relatively uniform response, for daylight hours. Figure 16 shows the dependence of $CH_4$ AKs on SZA and $CH_4$ total column, with sensitivity to surface $CH_4$ that increases with increasing values, especially for SZA, as expected from Figure 15.

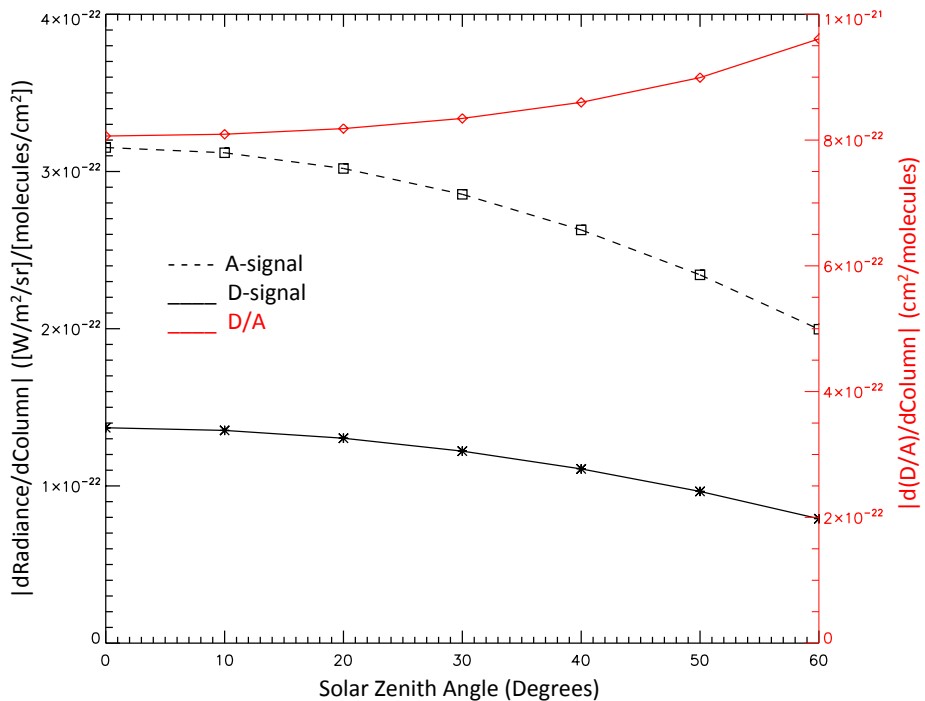

**Figure 15:** Variation of CHRONOS sensitivity to surface $CH_4$ with solar zenith angle (SZA). Absolute values for the surface layer $CH_4$ Jacobians are plotted for the A-signal, D-signal and D/A ratio as a function of SZA.

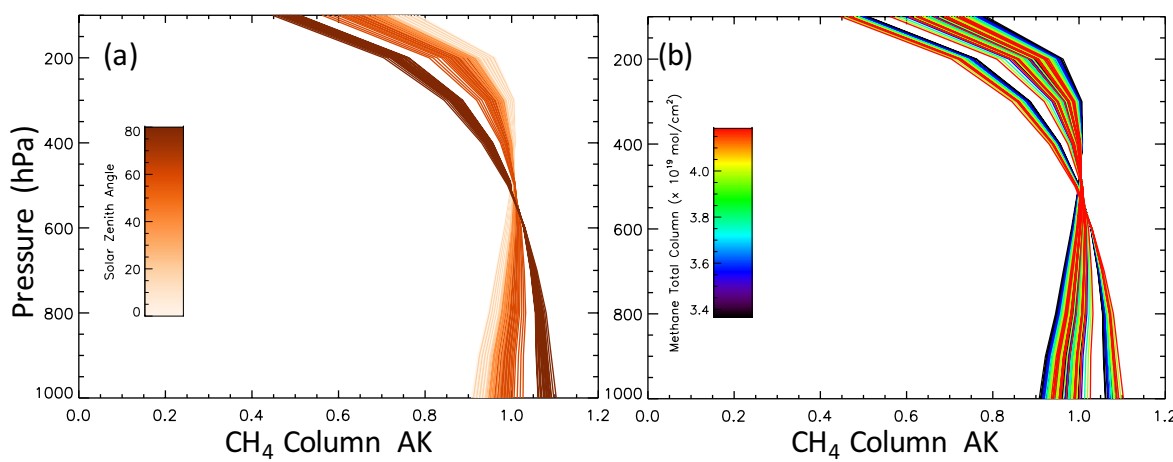

**Figure 16:** Variation of CHRONOS $CH_4$ averaging kernels for different SZA (left) and $CH_4$ total column (right) for 100 simulations with different SZA and $CH_4$ column values. Within each "group" of SZA values, the lesser AK dependence on low-to-high values of $CH_4$ can be seen.

## 6 Relationship of CHRONOS to Current and Future Missions

CHRONOS addresses key NASA science goals and National Research Council Decadal Survey questions with heritage from past satellite instruments and opportunities for synergistic observations with current and upcoming LEO and GEO platforms. In particular, CHRONOS would complement planned NASA missions for air quality and carbon cycle science. It would deliver air pollutant measurements identified in the 2007 Decadal Survey GEO-CAPE mission (NRC, 2007) and address currently unmet science objectives described in the GEO-CAPE Science Traceability Matrix (Fishman et al., 2012).

### 6.1 Other Satellite CO Observations

Along with MOPITT, satellite measurements of CO in the MWIR (4.6 μm) are available from AIRS, a grating spectrometer on EOS-Aqua launched in 2002 (Auman et al., 2003); the IASI FTS instruments on MetOp-A, B and C, launched in 2006, 2012 and expected in 2018 (Crevoisier et al., 2014); and the CrIS FTS instruments on Suomi NPP, launched 2011, and JPSS1-4 with projected launches starting in 2017 (Gambacorta et al., 2014). These LEO observations give daily global coverage at morning (IASI) and afternoon (AIRS, CrIS) equator crossings with sensitivity to CO in the middle troposphere for most observing conditions (George et al., 2009). The measurements are expected to be available during the CHRONOS mission time frame, and will provide valuable intercomparisons for the MWIR CHRONOS CO channel.

TROPOMI, a UV-VIS-NIR-SWIR spectrometer, launched in October 2017, provides daily global coverage from LEO at a 13:30 equator crossing with 7 km x 7 km spatial resolution and 10% column precision and 15% accuracy for SWIR (2.3 μm) CO observations (Veefkind et al., 2012). The TROPOMI CO measurements will provide true total column CO retrievals with more spatial coverage than MOPITT, but will not have MOPITT's CO vertical profile information. GOSAT-2 (http://www.gosat2.nies.go.jp), with expected launch in 2018, will also measure SWIR CO bands but with measurements spaced around 200 km apart and large gaps in the ground sampling. CHRONOS multispectral CO measurements could provide vertical profiles of CO over the continental U.S. domain every 10 minutes, along with total column CO that can be compared to TROPOMI and GOSAT-2. The LEO observations of CO outside of the CHRONOS field of regard would be useful for constraining CO transported from sources outside North America.

NASA selected the GeoCARB mission in November 2016, with capability to measure CO in one spectral region (Polonsky et al. 2014; Kumer et al., 2013) and primary carbon cycle science objectives unrelated to air pollution transport. Compared to the CHRONOS requirement for CO measurement in two spectral regions, this GeoCARB limitation to CO in one spectral region precludes GeoCARB from evaluating vertical pollution transport, or providing the test of these

atmospheric motions as calculated by models (NAS, 2017). Both Polonsky et al. (2104) and Kumer et al. (2013) describe mission descopes that eliminate GeoCARB measurements of CO entirely if needed to ensure success for GeoCARB $CO_2$ and solar induced fluorescence science objectives.

**6.2 Other Satellite $CH_4$ Observations**

GOSAT, launched in 2009, measures $CH_4$ from LEO in the SWIR (1.6 μm), with relatively sparse

coverage, a 10-km diameter footprint and column precision around 0.6% for single observations (Schepers et al., 2012). Improved sampling, coverage and precision are expected for GOSAT-2. Wecht et al. (2014a) show that hourly GEO SWIR $CH_4$ observations over California with 4 km x 4 km spatial resolution and 1.1% precision provide about 20 times the information for estimating $CH_4$ emissions compared to 3 days of GOSAT observations. This means that one 10-minute

collection of CHRONOS data would provide more information than a year of GOSAT observations, assuming $1/\sqrt{N}$ improvement for 365 days. Turner et al. (2015) used 3 years (2009-2011) of GOSAT $CH_4$ measurements to estimate North American emissions with 1/2° x 2/3° (~50 km x 70 km) spatial resolution, and found significant differences with the a priori inventory for anthropogenic emissions. Assuming the same information scaling found by Wecht et al. (2014a),

CHRONOS would be able to quantify $CH_4$ emissions for this spatial scale on a daily basis, with the capability to assess more rapid emission changes for events such as the 2015 Aliso Canyon gas leak (Conley et al., 2016).

TROPOMI in LEO uses near infrared (NIR) 0.76 μm and SWIR (2.3 μm) bands for $CH_4$ measurements and has an expected 0.6 % precision for single column $CH_4$ retrievals at 7 km x 7

km spatial resolution (Butz et al., 2012). Based on an analysis in Jacob et al. (2016), TROPOMI should be capable of regional scale quantification of $CH_4$ emissions. The daily probability of viewing sources that are either transient or obscured by clouds would be higher for CHRONOS in GEO than for TROPOMI, since CHRONOS could observe the entire continental U.S. domain six

times during each daylight hour. CHRONOS also has a higher probability of cloud-free observations given its smaller pixel size (see Figure 11).

GeoCARB describes $CH_4$ measurements in the SWIR (2.3 μm) region with 1% precision three times per day at 5 km x 5 km spatial resolution (O'Brien et al., 2016), although earlier studies (Kumer et al., 2013) explored $CH_4$ measurements at 1.65 μm. GeoCARB's more frequent $CH_4$ observations than TROPOMI may provide for similar precision in a smaller spatial footprint than TROPOMI. CHRONOS could observe $CH_4$ as often as every 10 minutes in daylight with 0.7% precision and 4 km x 4 km resolution. These frequent CHRONOS $CH_4$ measurements could be co-added to improve hourly precision, or used to examine anthropogenic source evolution over time.

For emissions on a 1/2° x 2/3° grid, Wecht et al. (2014a) show that GEO-CAPE SWIR $CH_4$ hourly observations (assuming 1.1% column precision) have 2.4 times the information of daily TROPOMI for estimating $CH_4$ emissions. More work is needed using OSSEs to understand how to optimally exploit LEO observations of $CH_4$ and CO, especially from TROPOMI and GOSAT-2, in combination with the information on diurnal variability that CHRONOS could provide. This extends to examination of the North American carbon budget since CO and $CH_4$ measurements from CHRONOS, in conjunction with detailed $CO_2$ observations from planned and operating missions, would allow differentiation of anthropogenic combustion and wildfire sources of $CO_2$.

Missions with the ability to measure "true" columns for CO and/or $CH_4$ (i.e., using SWIR spectra for the measurement) are summarized in Table 3. Note that for CHRONOS, 10% precision on CO observations meets the GEO-CAPE CO precision requirement of 10 ppbv. The CHRONOS 0.7% precision for $CH_4$ observations is achieved in a single 9.7-minute data collection; improved precision can be achieved by combining multiple data collections.

**Table 3: Relationship of CHRONOS to current and future CO and $CH_4$ missions. CHRONOS contributes unique observations of multispectral CO for tracing air pollution transport, and temporally dense $CH_4$ observations to improve emissions estimates across a continental domain.**

| Instrument | MOPITT | TROPOMI | TANSO-FTS-2 | Sentinel-5/UVNS | GeoCARB | CHRONOS |
|---|---|---|---|---|---|---|
| Instrument type | Gas Filter Correlation Radiometer | Grating Spectrometer | Fourier Transform Spectrometer | Grating Spectrometer | Grating Spectrometer | Gas Filter Correlation Radiometer |
| Spacecraft | NASA Terra | ESA Sentinel-5P | GOSAT-2 | METOP SG A1 | Commercial | Proposed |
| Launch Date | 1999 | 2017 | 2018 | 2021 | 2022 | NET 2024 |
| Source of Info | (Drummond, et al., 2010) | ATBD, (Veefkind, et al., 2012) | (Matsunaga et al., 2017) | (Ingmann, et al., 2012) | (O'Brien, et al., 2016) | This Work |
| Orbit | LEO SSO | LEO SSO | LEO SSO | LEO SSO | GEO | GEO |
| Domain | Near Global | Near Global | Near gobal | Near Global | North/South America | North America |
| Pixel Size, km$^2$ | 22 x 22 | 7 x 7 | 9.7 x 9.7 | 7.5 x 7.5 | 5 x 5 | 4 x 4 |
| Revisit | 3 days | Daily | 6 days | Daily | 3x/day | Sub-hourly* |
| $CH_4$ Spectral Region, µm | 2.222-2.293 | 2.303-2.385 | 1.563-1.695 5.56-8.45 | 1.590-1.675 | 2.301-2.346 | 2.250-2.313 |
| $CH_4$ Column Precision, % | - | 0.6 | 0.6 | 1 | 0.6 | 0.7 |
| CO Spectral Regions, µm | 2.323-2.345 | 2.303-2.385 | 1.923-2.381 | 2.305-2.385 | 2.301-2.346 | 2.313-2.364 |
| | 4.562-4.673 | - | - | - | - | 4.562-4.673 |
| CO Column Precision, % | 10 | 10 | 10 | 10 | 10 | 10 |

**\* up to 6 observations per hour**

## 7 Conclusions

We report a new capability for space based measurements of the important air pollutants carbon monoxide (CO) and methane ($CH_4$) as often as six times per hour. CO and $CH_4$ abundance are chemically linked in Earth's atmosphere as the principal sinks of hydroxyl. Sub-hourly observations of CO abundance, which is highly variable in space and time, can reveal new knowledge of the vertical and horizontal transport of air pollution. When sub-hourly observations of more slowly varying $CH_4$ abundance are combined, the temporally dense observations can significantly improve the precision of $CH_4$ emissions estimates. Observing System Simulation Experiments reported elsewhere show that improved CO and $CH_4$ emissions estimates can improve air quality forecasts that protect public health.

The CHRONOS investigation using 2-D imaging with full spectral resolution, would contribute the only sub-hourly CO and $CH_4$ observations for the U.S. component of an international GEO satellite constellation for atmospheric composition (CEOS, 2011) that includes the ESA/EUMETSAT Sentinel 4 mission over Europe and the Korean MP-GEOSAT/GEMS over Asia, along with the NASA TEMPO mission. LEO components (Sentinel 5/UVNS, TROPOMI,
GOSAT-2) of the constellation provide the global context (Table 3) for CHRONOS observations in assessing regional-to-global emissions and transport.

The main points defining the CHRONOS science investigation may be summarized as follows:

1. CHRONOS would deliver the first sub-hourly observation capability for comprehensive U.S. $CH_4$ and CO emission inventories and the ability to distinguish local from transported
air pollution.
2. At the county scale, CHRONOS would enable new estimates of rapidly changing, highly variable $CH_4$ and CO emissions from growing natural gas extraction and increasingly frequent and severe wildfires. These emissions estimates are essential for air quality, climate, and energy management decisions.
3. The dense data from sub-hourly air pollution observations at fine spatial resolution (nominally 4 km × 4 km) over the entire greater North American domain would quantify diurnal changes in air pollution and discriminate different source regions for urban and rural emission activities.
4. CHRONOS' multispectral CO retrieval would provide vertical information near the
surface in addition to the free troposphere to distinguish local air pollution from transported air pollution through horizontal and vertical tracking.
5. CHRONOS observations would strengthen the international air quality satellite constellation.

These science goals would be achieved by taking advantage of a simple, low-risk instrument
design that is well suited to the CHRONOS CO and $CH_4$ measurements. The GCFR heritage follows the successful 17-year, on-orbit operation of MOPITT over a wide range of observing conditions. This technique provides for high effective spectral resolution for the target gases, high signal levels compared to other types of spectrometers with similar spectral sensitivity, and small impact from signals due to interfering gases, aerosols, clouds and changing scene.

**Acknowledgements**

This work was partly supported by NASA grant NNX15AK98G. The National Center for Atmospheric Research (NCAR) is sponsored by the National Science Foundation. The NCAR MOPITT project is supported by the NASA Earth Observing System Program. We thank Glenn Diskin and the DACOM measurement team at NASA Langley for providing the DISCOVER-AQ $CH_4$ measurements shown in Figure 3.

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
