# Peer review of "The CHRONOS mission: Capability for sub-hourly synoptic observations of carbon monoxide and methane to quantify emissions and transport of air pollution"

_Atmospheric Measurement Techniques, 2017_

## Referee Comment (RC1) · Anonymous Referee #1 · 27 Jul 2017

<General Comments>

$CO_2$ and $CH_4$ monitoring with gas filter correlation technology from GEO is very important mission from both global warming and air quality monitoring points of view. Observation needs are well described. Recently many GEO and LEO GHG monitoring programs have been proposed. The authors should describe difference from the Geo-CARB program using grating spectrometer technology. CHRONOS has advantage to measure both solar reflected light from surface and thermal radiation from middle of the troposphere. However, it is not clear gas filter correlation technique is more accurate and/or precise than other technique such as grating spectrometer and FTS in CH4 retrieval. How to achieve 1% accuracy in CH4 retrieval under aerosol and high thin cloud condition without light path modification information should be described in more detail. Authors mention single case of aerosol but thin cloud such as high altitude cirrus is not discussed. Authors proposed use of GOES satellite data for cloud detection but aerosol and thin clouds are difficult to filter out. Major revision is needed.

<Specific Comments>

(1) Plumes Page 5, Fig 1 Description of diurnal variation of CO emission and typical wind speed in WRF-Chem will help readers' understanding Page 10, Fig. 3 Description of CH4 emission source in Greeley, CO will help readers' understanding.

(2) Page 7, Line 162, " âĹij6 – 12% " It is not clear. Does it mean between 6 and 12%?

(3) Page 10, Line 242, "Air quality criteria to protect public health" Reference or explanation is needed.

(4) Page 12, Line 298 The brief description of the reason why $5\mu$rad is needed.

(5) Page 13, Line 313, "the effect of variations in the underling surface" Does it mean fine spectral structure of surface albedo?

(6) Page 15, Figure 6, "solid red lines at filter half-power point" Is it 50% transmittance point? The transmittance at red line looks about 40%.

(7) Page 16, Line 366 (<10%) Accuracy requirement for CO and CH4 must be different but instrument is similar. CO accuracy of 10% is reasonable and was demonstrated with MOPIT. How is the accuracy of 1% achieved in the CH4 retrieval? Aerosol and thin cloud cause bias error and averaging cannot reduce the bias. Recent CH4 satellite retrieval such as GOSAT use O2A band in 0.76 micron to estimate light path modification by aerosol and CH4.

(8) Page 17, Lines 375-333, "there 3 minute retrieval" "These 3 minute retrieval" and

relation between âĹij3 min intervals and retrievals are not clear. What is the definition of "single (âĹij10 min) data"?

(9) Page 21, Line 455, "all digital" What do the authors mean by "all digital"? Usually detectors and readout electronics have analogue portion such as amplifier and analogue to digital converter.

(10) Page 22, Line 487, "radiance calibration" Brief description of radiance calibration is needed.

(11) Page 23, Figure 11, vertical axis "#obs in domain/# pixels Explanation is needed.

(12) Page 30, Line 639, "launch in 2017" Page 32 table 3 OCO-3 (2017-) I think GOSAT-2 launch is scheduled to be in 2018 as the authors indicated in Table 3. I think OCO-3 has less possibility to be launched this year.

<Technical Corrections>

(1) Page 24, Line 522, "total hydrometeors > 10-8/kg/kg" Is it 10ˆ-8?

(2) Page 34, Line 723, "et al." and many others. AMT authors guideline says "Please supply the full author list with last name followed by initials." Other formats also do not meet the guideline.

---

## Referee Comment (RC2) · Anonymous Referee #2 · 18 Sep 2017

General comments:

The manuscript 'The CHRONOS mission: Capability for sub-hourly synoptic observations of carbon monoxide and methane to quantify emissions and transport of air pollution' by D. P Edwards at al. describes a new mission concept of satellite remote sensing of both trace gases using a geostationary orbit. The proposed instrument is based on MOPITT instrument heritage. Although having such a mission would provide exciting new measurements, the paper itself provides only little scientific news. The MOPITT heritage is discussed extensively in the literature and the possibility to

observe CO and CH4 using a geostationary orbit is already discussed e.g. by Butz et al., 2015 and O'Brein et al., 2016. Furthermore, the downstream from mission objectives to instrument and product requirements is not always traceable for me. Also from a technical point of view, the paper requires major revision. For example, Sec 2.2 formulates the science objectives and already concludes that CHRONOS meets all the objectives although the instrument, its spectral coverage and product accuracy and precision is discussed much later in the paper. Moreover, the science objectives for CH4 geo observations are not always convincing to me. The science objectives of the manuscript are mainly driven by CO. For methane, climate and emission monitoring relevant question are mentioned but the need for a geo orbit for this purpose is not discussed sufficiently. For example, figure 2 suggests that on hourly time scales only CO measurements provide relevant information and so one may conclude that for this time scale CH4 measurements are of limited use. In section 2.3, the impact of CH4 on air quality is mentioned via its reaction with OH. The expected CHRONOS CH4 precision and accuracy are provided in Tab 2 and meet typical requirements for climate and emission monitoring, however, it is not clear to me if air quality forecast can really be improved with these uncertainties.

Specific comments:

1. Figure 1: What is the spatial coverage of the MOPITT panels? For a better comparison, both MOPITT and WRF-Chem images should use the same color code, which should be indicated in a corresponding figure legend.

2. Figure 3: The figure shows CH4 in situ measurements with significant enhancements whereas changes in the total column is much less (4.9 %). I think this enhancement is given on the spatial sampling of the in situ measurements but do not necessarily represent the CH4 enhancement for a 4x4 km2 sampling of CHRONOS. I can imagine this makes a different.

3. Figure 4 and 12: These figures do not present new material and can be discussed

in the text with appropriate references.

4. Page 23, line 509: To my knowledge, it is not demonstrated that CH4 can be retrieved from real MOPITT data with a cloud coverage of 5 %. I doubt that this is possible considering the strict cloud filtering of GOSAT observations for CH4 retrieval.

5. Page 19 line 414: The aerosol optical depth should be provided at a reference wavelength within the SWIR fit window. Depending on the size of the aerosol parameter, this can be very small.

6. Table 2: I think, a discussion for elevated aerosol layers and cirrus is needed for a better error estimate. From other missions, we know that these are relevant error sources.

7. Table 2: Here a precision requirement of <10 % is given, whereas Fig 2 indicates that urban air quality daily evolution is in the order of 1-2 ppm. I doubt that with this large precision, urban daily evolution can be measured. See also page 17, line 383-385.

8. Page 15 line 335-340. The SCIAMACHY CH4 product is inferred from 1.6 micron measurements, GOSAT also uses the methane sensitivity at 1.6 micron, which in both cases differ from the CHRONOS SWIR window at 2.2 micron.

9. Section 6: I am not sure if I overlooked it, but when discussing synergies with other mission an indication of a launch window is required. I think, also the Sentinel-5 mission and IASI-NG should be mentioned here. I also miss a discussion of GEOCarb, which would measure CO, CO2 and CH4. Because the mission concept is already published, it should not be ignored in this manuscript.

10. Table 3: This table does not provide new information, which is not already discussed in the text. It also does not fit the format of a science publication to my opinion.

---

## Author Comment (AC1) · 28 Oct 2017

The comment was uploaded in the form of a supplement:
https://www.atmos-meas-tech-discuss.net/amt-2017-194/amt-2017-194-AC1-supplement.pdf
* * *

---

## Author Response (AR1)

**amt-2017-194**

**The CHRONOS mission: Capability for sub-hourly synoptic observations of carbon**
**monoxide and methane to quantify emissions and transport of air pollution**

**David P. Edwards, Helen M. Worden, Doreen Neil, Gene Francis, Tim Valle, and Avelino**
**F. Arellano Jr.**

**Response to Reviewer 1:**

We thank the reviewer for their careful evaluation of our manuscript. We address each comment
(in blue) with an embedded response (in black) below. We detail new text that has been added to
the revised manuscript (in green).

General Comments:

CO2 and CH4 monitoring with gas filter correlation technology from GEO is very important
mission from both global warming and air quality monitoring points of view. Observation needs
are well described. Recently many GEO and LEO GHG monitoring programs have been
proposed. The authors should describe difference from the Geo-CARB program using grating
spectrometer technology.

We agree that more description of the differences between CHRONOS and GeoCARB, which
was recently selected for the NASA EVM-2 program, is needed. We have added the following
text to Section 6.1:

NASA selected the GeoCARB mission in November 2016, with capability to measure CO in one
spectral region (Polonsky et al. 2014; Kumer et al., 2013) and primary carbon cycle science
objectives unrelated to air pollution transport. Compared to the CHRONOS requirement for CO
measurement in two spectral regions, this GeoCARB limitation to CO in one spectral region
precludes GeoCARB from evaluating vertical pollution transport, or providing the test of these
atmospheric motions as calculated by models (NAS, 2017). Both Polonsky et al. (2104) and
Kumer et al. (2013) describe mission descopes that eliminate GeoCARB measurements of CO
entirely if needed to ensure success for GeoCARB $CO_2$ and solar induced fluorescence science
objectives.

And to Section 6.2:

GeoCARB describes $CH_4$ measurements in the SWIR (2.3 μm) region with 1% precision three
times per day at 5 km x 5 km spatial resolution (O'Brien et al., 2016), although earlier studies
(Kumer et al., 2013) explored methane measurements at 1.65 μm. GeoCARB's more frequent
methane observations than TROPOMI may provide for similar precision in a smaller spatial
footprint than TROPOMI.  CHRONOS could observe $CH_4$ as often as every 10 minutes in
daylight with 0.7% precision and 4 km x 4 km resolution. These frequent CHRONOS $CH_4$
measurements could be co-added to improve hourly precision, or used to examine anthropogenic
source evolution over time.

GeoCARB parameters are also included in Table 3, now revised in response to Reviewer 2.

CHRONOS has advantage to measure both solar reflected light from surface and thermal radiation from middle of the troposphere. However, it is not clear gas filter correlation technique is more accurate and/or precise than other technique such as grating spectrometer and FTS in CH4 retrieval.

The gas filter correlation technique achieves accuracy and precision in trace gas retrieval similar to grating spectrometers and FTS by virtue of very high effective spectral resolution and high throughput (low noise). We have clarified the choice of spectral technique by adding to the discussion in Section 3.1:

The effective spectral resolution of the GFCR response function (Edwards et al., 1999, figure 3) matches the pressure-broadened Lorentz full-width-half-maximum (FWHM) for weak-absorption lines (Beer, 1992), and ranges from 0.08 cm$^{-1}$ to 0.16 cm$^{-1}$ for 200 hPa to 800 hPa GFCR gas cells (Pan et al., 1995). This optimal spectral resolution for measuring tropospheric trace gas absorption and for probing the spectral line profile to obtain information on the trace gas atmospheric vertical distribution is difficult to achieve for most spectrometers without sacrificing signal amplitude (grating spectrometers) or increasing noise (Fourier transform spectrometers). The limitation for the GFCR technique is that atmospheric retrievals are made only for those gases contained within the cells of the instrument. However, for observations of CO and CH$_4$ from GEO (50 times farther from Earth than LEO), the advantages of both fine spectral resolution and high throughput provided by CRONOS's gas filter correlation radiometry make for a particularly robust measurement approach.

New references:

Polonsky, I. N., O'Brien, D. M., Kumer, J. B. and O'Dell, C. W.: Performance of a geostationary mission, geoCARB, to measure CO2, CH4 and CO column-averaged concentrations. Atmospheric Measurement Techniques, 7(4), pp.959-981, 2014.

Kumer, J.J.B., Rairden, R.L., Roche, A.E., Chevallier, F., Rayner, P.J. and Moore, B.: September. Progress in development of Tropospheric Infrared Mapping Spectrometers (TIMS): GeoCARB Greenhouse Gas (GHG) application. In Infrared Remote Sensing and Instrumentation XXI (Vol. 8867, p. 88670K). International Society for Optics and Photonics, 2013.

National Academies of Sciences, Engineering, and Medicine: Powering Science: NASA's Large Strategic Science Missions. Washington, DC: The National Academies Press. https://doi.org/10.17226/24857, p33, p81, 2017.

O'Brien, D. M., Polonsky, I. N., Utembe, S. R., and Rayner, P. J.: Potential of a geostationary geoCARB mission to estimate surface emissions of CO$_2$, CH$_4$ and CO in a polluted urban environment: case study Shanghai, Atmospheric Measurement Techniques, 9, 4633–4654, https://doi.org/10.5194/amt-9-4633-2016, 2016.

Pan, L., Edwards, D. P., Gille, J. C., Smith, M. W., and Drummond, J. R.: Satellite remote sensing of tropospheric CO and CH4: forward model studies of the MOPITT instrument, *Appl. Opt.*, *34*(30), 6976–6988, doi:10.1364/AO.34.006976, 1995.

Beer, R.: Remote Sensing by Fourier Transform Spectrometry, *Wiley, New York*, 1992.

How to achieve 1% accuracy in CH4 retrieval under aerosol and high thin cloud condition
without light path modification information should be described in more detail.

For the retrieval of $CH_4$ in the presence of clouds and aerosols, we added to Section 3.2:

SCIAMACHY and GOSAT $CH_4$ SWIR retrievals are sensitive to scattering by dust, aerosols and
thin cirrus (Gloudemans et al., 2008; Schepers et al., 2012) and address these errors by using
$CO_2$ (with known abundance) as a proxy for the scattering effects or by performing a physical
retrieval of effective parameters for the scattering layer. For GOSAT $CH_4$ data, these two
approaches yield similar precision (~17 ppb) and biases less than 1% compared to TCCON
(Wunch et al., 2010), but with lower bias for the proxy method (Schepers et al., 2012). In the
proxy retrieval using $CO_2$, the dry mole fraction of $CH_4$ ($x_{CH4}$) is computed by $x_{CH4} =$
$\frac{[CH4]}{[CO2]} x_{CO2}$ where $[CH4]$ and $[CO2]$ are the retrieved columns from spectral radiances that are
close in wavenumber and $x_{CO2}$ is the dry mole fraction computed from a global model of
atmospheric $CO_2$ (Frankenberg et al., 2005; Schepers et al., 2012). This method assumes that
aerosol scattering modifies the light path for $CO_2$ and $CH_4$ spectral absorption in the same way,
and that model values for $x_{CO2}$ are accurate.

Retrievals with GFCR measurements are similar to the "proxy retrieval" but they correct the
input radiance instead of the retrieved column, and do not make assumptions about aerosol
scattering in different spectral bands or rely on knowing $CO_2$ abundance. CHRONOS uses the
D/A signal ratio where D and A are both modified in the same way by aerosol scattering, which
has a smooth spectral behavior over the CHRONOS bandpass. This ratio gives an accurate total
column amount, but to compute a dry mole fraction (xCH4), we require additional information
about the surface pressure (for example, from GOES-16 meteorological data) in order to estimate
the dry air column. In general, GFCR retrievals are more resilient than spectral radiance
measurements to errors in surface and contaminant species assumptions due to the use of
radiance differences and ratios (Pan et al., 1995).

Authors mention single case of aerosol but thin cloud such as high-altitude cirrus is not
discussed. Authors proposed use of GOES satellite data for cloud detection but aerosol and thin
clouds are difficult to filter out.

As described in Section 4, CHRONOS's primary cloud detection comes through its own GFCR
measurements based on many years of experience with MOPITT cloud detection. The fact that
CHRONOS is in GEO and making observations of the same scene sub-hourly, also affords some
advantages for cloud detection by means of being able to look at very frequent signal differences
in combination with GEO imagery from GOES-16 ABI. We have added the following text to
Sec. 4:

While the approach of using D/A for retrievals discussed in Section 3.3 will cancel some of the
errors due to undetected aerosols or clouds (e.g., thin cirrus), remaining retrievals errors (e.g.,
O'Dell et al., 2011), particularly for $CH_4$, will require further study using both CHRONOS
radiances and GOES-16 ABI observations.

**Specific Comments**

(1) Plumes Page 5, Fig 1 Description of diurnal variation of CO emission and typical wind speed
in WRF-Chem will help readers' understanding Page 10, Fig. 3 Description of CH4 emission
source in Greeley, CO will help readers' understanding.

Clarified: The text of the Figure 1 caption has been updated to state that the WRF-Chem run is
driven by analyzed meteorology, and that changes in the distribution of CO are expected as a
result of changes in both emissions and meteorology.

**Figure 1:** Comparison of MOPITT and CHRONOS spatial and temporal coverage over a 5-hour
period. The top panels show MOPITT retrievals of near-surface CO for Tuesday Aug. 1, 2006,
with pink colors indicating low CO ($\sim$ 60 ppbV) and green to red indicating higher values (200 –
300 ppbV). The middle and bottom panels show a simulation of CHRONOS observations using
WRF-Chem (Grell et al., 2005) at 4 km horizontal resolution driven by analyzed meteorology
(Barth et al., 2012) for the same date. Here blue colors indicate low CO ($\sim$60 ppbV), red colors
indicate high CO ($\sim$300 ppbV) and light greys indicate clouds. Circled areas in the zoomed
bottom panels provide detailed examples of changes in CO concentrations over the 5-hour period
with pollution from Chicago moving to the west and clouds moving east over the Washington
DC area. Urban traffic patterns and weather fronts change the distribution of air pollution
throughout the day. Sub-hourly CHRONOS data could assist with attributing the sources of
pollution and determining areas affected downwind.

New reference added:

Barth, M. C., Lee, J., Hodzic, A., Pfister, G., Skamarock, W. C., Worden, J., Wong, J., and
Noone, D.: Thunderstorms and upper troposphere chemistry during the early stages of the 2006
North American Monsoon, *Atmos. Chem. Phys.*, **12**, 11,003-11,026, doi:10.5194/acp-12-11003-
2012, 2012.

Similarly, the text of the Figure 3 caption now includes source description:

**Figure 3:** Aircraft in situ measurements of $CH_4$ from the FRAPPE-DISCOVER-AQ in the
Colorado Front Range on Aug. 2, 2014. Vertical profiles were measured over cities, identified by
spiral flight tracks (each spiral has $\sim$10 km radius).  Note that the highest values of $CH_4$ are
plotted last. Total column $CH_4$ computed from the vertical profiles is different by 4.9% between
Ft. Collins (urban) and Greeley (oil/gas and feedlot operations). CHRONOS spatial resolution is
indicated by the overlaid grid, illustrating that CHRONOS column measurements would have the
spatial resolution and precision to distinguish sub-hourly differences in county-
scale $CH_4$ abundances from space. Data courtesy of Glenn Diskin, NASA.

(2) Page 7, Line 162, It is not clear. Does it mean between 6 and 12%?

Text changed to read "…. between 6 and 12%.".

(3) Page 10, Line 242, "Air quality criteria to protect public health" Reference or explanation is
needed.

Clarified: The text referred to has been rewritten as: Nine months before the U.S. Environmental
Protection Agency was founded, air quality criteria were established for carbon monoxide (U.S.,

1970) to protect public health in compliance with the 1967 amendments (Public Law 90-148) to
the Clean Air Act of 1963 (Public Law 88-206).

(4) Page 12, Line 298 The brief description of the reason why 5μrad is needed.

The text "The displacement between a single paired gas/vacuum measurement is limited to ≤5
μrad/60 msec to ensure acceptable changes in ground pixel reflectance based on MOPITT
experience (Deeter et al., 2011), and on simulated radiance errors using representative GEO
spacecraft pointing data", has been rewritten to read:

Observation simulation studies using representative GEO spacecraft pointing data have been
performed to determine the effect of 'jitter' in spacecraft pointing during the acquisition of a
signal pair. The displacement between a single paired gas/vacuum measurement is limited to ≤5
μrad to ensure acceptable changes in ground pixel reflectance based on MOPITT experience
(Deeter et al., 2011). This requirement corresponds with a gas cell-to-vacuum cell frame time
limited to 60 msec, readily achievable with a physically realistic cell size and rotation frequency,
frame acquisition and readout rate. The large (>3000 kg) size of a commercial communications
spacecraft therefore serves to naturally attenuate jitter sources over very short time frames,
avoiding the need for a costly image stabilization subsystem.

(5) Page 13, Line 313, "the effect of variations in the underling surface" Does it mean fine
spectral structure of surface albedo?

Clarified: The text "the effect of variations in the underling surface" has been changed to read
"the effects of variations in the underlying surface temperature, emission, and reflectivity".

(6) Page 15, Figure 6, "solid red lines at filter half-power point" Is it 50% transmittance point?
The transmittance at red line looks about 40%.

These are the 50% transmittance points, now noted in figure caption.

(7) Page 16, Line 366 (<10%) Accuracy requirement for CO and CH4 must be different but
instrument is similar. CO accuracy of 10% is reasonable and was demonstrated with MOPIT.
How is the accuracy of 1% achieved in the CH4 retrieval? Aerosol and thin cloud cause bias
error and averaging cannot reduce the bias. Recent CH4 satellite retrieval such as GOSAT use
O2A band in 0.76 micron to estimate light path modification by aerosol and CH4.

The measurement accuracy requirements of the observations are set by the product accuracy
required to answer the science questions (multispectral CO accuracy 10%, and $CH_4$ accuracy 1%
as stated by the Reviewer). Measurement accuracy requirements are discussed in Section 3.2.
While the instrument is the same, the measurements of CO and CH4 and the underlying spectral
signatures and radiative transfer are different. The CHRONOS instrument acquires fewer or
additional observations in each spectral channel to achieve the required signal-to-noise. In
Section 3.3, Table1 provides the measurement passbands for optimized spectral sensitivity. We
have added to Table 1 the minimum signal-to-noise ratio for each measurement, and the number of observations needed to achieve that minimum SNR, and supplemented the text preceding the
Table as follows:

Table 1 lists the modeled signal-to-noise (SNR) and the total number of individual data
acquisitions in each pixel in the 2D detector array ("frames") obtained in a single 9.7-minute data
acquisition period, for the minimum radiance case defined from MOPITT on-orbit radiance
records. This minimum SNR provides at least 30% margin for meeting the radiance precision
requirements.

(8) Page 17, Lines 375-333, "there 3 minute retrieval" "These 3 minute retrieval" and relation
between âLij3 min intervals and retrievals are not clear. What is the definition of "single (âLij10
min) data"?

The original text appears to be corrupted. We have rewritten the text to clarify as follows:

Profile or column retrieval precision requirements are achieved in ground processing by
averaging geo-located, cloud screened radiances for three minutes (375 separate gas-vacuum
measurements for each product: CO [4.6 μm, 800 hPa], CO [4.6 μm, 200 hPa], CO [2.3 μm, 100
hPa]; and 750 measurements of $CH_4$ [2.2 μm, 800 hPa]). A single retrieval for each product is
performed on these averaged radiances. The process of averaging radiances and then retrieving
products is repeated for all data acquired in the 9.7-minute data acquisition period.

(9) Page 21, Line 455, "all digital" What do the authors mean by "all digital"? Usually detectors
and readout electronics have analogue portion such as amplifier and analogue to digital
converter.

The "all-digital" focal plane arrays became available for science use in the early 2000s. For all of
the cited arrays, signal amplification and analog-to-digital conversion occur in the readout
integrated circuit (ROIC) at each pixel, leading to the term "in-pixel digitization" or "all
digital". This type of array is what enables CHRONOS to quantify very small differences in
radiance. We have added a reference to:

Brown, M.G., Baker, J., Colonero, C., Costa, J., Gardner, T., Kelly. M., Schultz, K., Tyrrell, B.,
and Wey, J.: Digital-pixel focal plane array development, Proc. SPIE 7608, Quantum Sensing
and Nanophotonic Devices VII, 76082H (January 22, 2010); doi:10.1117/12.838314, 2010.

Although the title above says "digital pixel", text in this and other papers refer to "all digital" or
just "digital" focal plane arrays, which is now a common usage we adopt in the manuscript.

(10) Page 22, Line 487, "radiance calibration" Brief description of radiance calibration is
needed.

We have added a brief description of radiance calibration to Section 4 as follows:

For on-orbit radiance calibration, CHRONOS views high-precision hot and cold black bodies
and deep space for the MWIR channels, and a tungsten lamp (LandSat Operational Land Imager
heritage) and a closed aperture for the SWIR calibration within each 10-minute data acquisition.

(11) Page 23, Figure 11, vertical axis "#obs in domain/# pixels Explanation is needed.

Clarified: Added text to the figure caption: "#obs in domain/# pixels (the number of cloud-free
pixels observed as a fraction of the total number of pixels in the region)".

We have changed the GOSAT-2 launch to 2018 at this location and in Table 3.

Page 32 table 3 OCO-3 (2017-) I think GOSAT-2 launch is scheduled to be in 2018 as the
authors indicated in Table 3. I think OCO-3 has less possibility to be launched this year.

Table 3 has been revised in response to Reviewer 2.

Technical Corrections

Corrected: Changed to $10^{-8}$/kg/kg.

(2) Page 34, Line 723, "et al." and many others. AMT authors guideline says "Please supply the
full author list with last name followed by initials." Other formats also do not meet the guideline.

Corrected: Formats have been changed to match guidelines throughout.

**amt-2017-194**

**The CHRONOS mission: Capability for sub-hourly synoptic observations of carbon**
**monoxide and methane to quantify emissions and transport of air pollution**

**David P. Edwards, Helen M. Worden, Doreen Neil, Gene Francis, Tim Valle, and Avelino**
**F. Arellano Jr.**

**Response to Reviewer 2:**

We thank the reviewer for their careful evaluation of our manuscript. We address each comment
(in blue) with an embedded response (in black) below. We detail new text that has been added to
the revised manuscript (in green).

General comments:

The manuscript 'The CHRONOS mission: Capability for sub-hourly synoptic observations of
carbon monoxide and methane to quantify emissions and transport of air pollution' by D. P
Edwards at al. describes a new mission concept of satellite remote sensing of both trace gases
using a geostationary orbit. The proposed instrument is based on MOPITT instrument heritage.
Although having such a mission would provide exciting new measurements, the paper itself
provides only little scientific news. The MOPITT heritage is discussed extensively in the
literature and the possibility to observe CO and CH4 using a geostationary orbit is already
discussed e.g. by Butz et al., 2015 and O'Brein et al., 2016.

We respectfully disagree with the comment "the paper itself provides only little scientific
news… and the possibility to observe CO and CH4 using a geostationary orbit is already
discussed e.g. by Butz et al., 2015 and O'Brein et al., 2016."

The CHRONOS mission concept addresses tropospheric (air pollution) chemistry, specifically,
carbon monoxide and methane, the two principal sinks for the hydroxyl radical.  Hydroxyl
provides the ability of Earth's troposphere to cleanse itself of trace constituents that are harmful
and even toxic to plants, animals, and people. When combined with NASA's planned TEMPO
observations, CHRONOS observations meet the tropospheric chemistry science objectives of the
GEO-CAPE mission (Fishman et al., 2012).

The CHRONOS focus on tropospheric chemistry then results in requirements for frequent time
sampling (sub-hourly), and for simultaneous sampling over extended domains ("snapshot") that
would be significantly impaired by instrument solutions that take hours to scan a continental
domain and sampling patterns that are limited to few times daily, as described in O'Brien et al.
(2016) or Butz et al. (2015).  An instrument capable of instantaneous observations everywhere
over a continental domain with the ability to provide full precision observations everywhere in
that domain within 10 minutes is scientific news for meeting the tropospheric chemistry
objectives.

Compared to the CHRONOS requirement for CO measurement in two spectral regions, the
GeoCARB limitation to CO in one spectral region means that GeoCARB would not able to
evaluate vertical pollution transport, or to provide the test of these atmospheric motions as calculated by advanced atmospheric models (NAS, 2017).  The Committee on Earth Observing
Satellites (CEOS) has identified the absence of multispectral CO observations after MOPITT as
a critical data gap when they met in June 2017.

We agree that MOPITT capabilities are well documented in the literature. An important point of
the paper is to report the rationale and specifics of the CHRONOS instrument evolution of
MOPITT (CHRONOS provides simultaneous, sub-hourly sampling everywhere in the domain
instead of temporally discontinuous orbital tracks with, at best, daily revisit; and CHRONOS
improves spatial resolution to 4 km x 4 km from MOPITT's 22 km x 22 km).

the downstream from mission objectives to instrument and product requirements is not always
traceable for me… Sec 2.2 …. already concludes that CHRONOS meets all the objectives
although the instrument …. is discussed much later in the paper

This paper was written with the intent of laying out the science case first and then describing the
measurement requirements and instrument. This may be different to some instrument papers, but
it is also becoming a frequent style for papers, reports, and surveys following the "Traceability
Matrix" approach (e.g., NRC, 2007, and the currently in-process NASA/NOAA/USGS 2017
Decadal Survey).

science objectives for CH4 geo observations are not always convincing to me

Section 6.2 discusses OSSEs by Wecht et al. (2014) that evaluated the potential of hourly
methane observations (such as we describe in this paper) for constraining emissions. Wecht et al.
report that hourly observations constrain methane emissions more than a factor of two better than
daily observations (TROPOMI), and significantly better than GOSAT.

it is not clear to me if air quality forecast can really be improved with these uncertainties

At the end of Section 2.2, we describe previously published OSSE studies that show the benefit,
compared to the ground-based monitoring network, of high spatial and temporal resolution
sampling from GEO in constraining transport patterns and in constraining the distribution of
near-surface CO concentrations. Specifically, Edwards et al (2009) cites improved skill scores
for near surface CO, and Zoogman et al (2014) demonstrates improvements to near surface
ozone from joint CO-ozone assimilations.

Specific comments:

1. Figure 1: What is the spatial coverage of the MOPITT panels? For a better comparison, both
MOPITT and WRF-Chem images should use the same color code, which should be indicated in
a corresponding figure legend.

Details of the MOPITT instrument are included in the references in the Introduction. MOPITT
uses a cross-track scan of 640 km that allows for almost complete coverage of the Earth's surface
in about 3 days, with pixels of 22 km x 22 km horizontal resolution. The intent of Figure 1 is to graphically depict spatial coverage as a function of hourly sampling for MOPITT compared to
what might be seen from CHRONOS. It is not intended as a comparison of MOPITT CO values
with the WRF-Chem model. Validation of MOPITT against both models and observations,
taking into account averaging kernel sensitivity and a priori assumptions not considered in this
figure, are covered by the MOPITT references in the manuscript (and references therein).

2. Figure 3: The figure shows CH4 in situ measurements with significant enhancements
whereas changes in the total column is much less (4.9 %). I think this enhancement is given
on the spatial sampling of the in situ measurements but do not necessarily represent the CH4
enhancement for a 4x4 km2 sampling of CHRONOS. I can imagine this makes a different.

While the individual in-situ measurements shown in Fig. 3 are indeed not at the CHRONOS
spatial resolution, the methane enhancements observed near oil/gas and feedlot operations during
the 2014 FRAPPE DISCOVER-AQ campaign were persistent over the spiral flight tracks shown,
each spiral with an approximate radius of 10 km (see e.g, Flynn et al., 2016). The total column
calculation that yields the quoted 4.9 % difference therefore corresponds to the CHRONOS pixel
scale. As the reviewer points out, high methane values at very small scale, as might be expected
from an individual pipe leak, would not probably not produce the necessary column enhancement
to be detected at the CHRONOS pixel scale. We are careful to suggest emissions estimates for
CH$_4$ at the county-level spatial scales (~40 km x 40 km) as demonstrated in the referenced OSSE
by Wecht et al. (2014a) as part of the studies for GEO-CAPE.

We clarify this in the revised Figure 3 caption: Vertical profiles were measured over cities,
identified by spiral flight tracks (each spiral has ~10 km radius).

3. Figure 4 and 12: These figures do not present new material and can be discussed in the text
with appropriate references.

While this material may be understood by some readers, these figures have never before been
published and we think they help elucidate the CHRONOS measurement concepts much more
clearly than we could describe with text. We have found frequently that audiences are not familiar
with the principles of GFCR, as compared to more usual grating spectrometer or FTIR
measurement approaches. CHRONOS also employs alternating gas/vacuum cells, which is
different to the MOPITT pressure-modulated and length-modulated GFCR technique. For these
reasons, we believe that Figure 4 provides valuable context.

Clarified: Specified 'the CHRONOS GFCR measurement' in the text of the Figure 4 caption.

A similar comment applies to Figure 12. Of all the current and planned CO measuring
instruments, MOPITT is the only one making multispectral CO retrievals. The other instruments
either make SWIR column measurements or TIR mid-troposphere measurements. As a result,
readers who are not familiar with the MOPITT literature will not appreciate the advantage of the
multispectral approach in providing independent near-surface CO concentrations to understand
emissions and pollution transport. This is a primary motivation for the CHRONOS concept.

4. Page 23, line 509: To my knowledge, it is not demonstrated that CH4 can be retrieved from
real MOPITT data with a cloud coverage of 5 %. I doubt that this is possible considering the strict cloud filtering of GOSAT observations for CH4 retrieval.

As outlined in Section 3.3 and references therein, MOPITT does not retrieve CH4 with or
without clouds, due to well-documented instrumental issues.

It is an advantage of GFCR that 'contaminating' signals that are spectrally flat across the
radiometer filter passband are effectively cancelled out by the D/A signal. We have added a
description to Section 4 as follows:

While the approach of using D/A for retrievals discussed in Section 3 will cancel some of the
errors due to undetected aerosols or clouds (e.g., thin cirrus), remaining retrievals errors (e.g.,
O'Dell et al., 2011), particularly for CH$_4$, will require further study using both CHRONOS
radiances and GOES-16 ABI observations.

5. Page 19 line 414: The aerosol optical depth should be provided at a reference wave- length
within the SWIR fit window. Depending on the size of the aerosol parameter, this can be very
small.

The value of AOD used in the 2.25 micron SWIR window aerosol simulation was 0.089. This is
now added to the text. This aerosol case was chosen to represent high pollution loading with the
most significant AOD values in our SWIR window.

We have added to Figure 7 caption: (...AOD is 0.089, which is obtained by scaling the OPAC
urban aerosol case by 1.5)

6. Table 2: I think, a discussion for elevated aerosol layers and cirrus is needed for a better
error estimate. From other missions, we know that these are relevant error sources.

We agree that this is a tricky problem and we expect to use the CHRONOS radiances as well as
GOES-16 ABI cloud measurements to diagnose and potentially flag observations that might not
be properly filtered by our cloud detection approach. We will also follow the approaches
identified for OCO-2 observations to detect and quantify retrieval errors due to undetected aerosol
and thin cirrus clouds (e.g., O'Dell et al., 2011). We have added discussions to Section 3.2 in
response to Reviewer 1 and to the cloud detection paragraph in Section 4 as noted in Specific
comment 4 above.

7. Table 2: Here a precision requirement of <10 % is given, whereas Fig 2 indicates that urban air
quality daily evolution is in the order of 1-2 ppm. I doubt that with this large precision, urban
daily evolution can be measured. See also page 17, line 383-385.

We recognize the opportunity for confusion between ppm and ppb and have corrected Figure 2
to show 1000-2000 ppb rather than 1-2 ppm for CO. Precision requirement of 10% is
approximately 10 ppb based on global average abundance of CO.

8. Page 15 line 335-340. The SCIAMACHY CH4 product is inferred from 1.6 micron
measurements, GOSAT also uses the methane sensitivity at 1.6 micron, which in both cases differ
from the CHRONOS SWIR window at 2.2 micron.

The measurement of CH4 at 2.2 microns was considered by SCIAMACHY prior to the detector
icing problem, is used by S5-P/TROPOMI, and will be used by GeoCARB. The spectral band
used for methane is summarized in the new Table 3.

9. Section 6: I am not sure if I overlooked it, but when discussing synergies with other
mission an indication of a launch window is required. I think, also the Sentinel-5 mission and
IASI-NG should be mentioned here.

We have revised Table 3 to reflect a CHRONOS launch no earlier than (NET) 2024 and have
included Sentinel-5. We chose not to include IASI-NG as it makes observations only in the
MWIR (similar to other sounders such as the current IASI instrument and CrIS). As described
in Sections 5.1 and 6.1, these instruments do not generally have measurement sensitivity to
the full column.

I also miss a discussion of GEOCarb, which would measure CO, CO2 and CH4. Because the
mission concept is already published, it should not be ignored in this manuscript.

We have added a discussion of GeoCARB in Section 6 and in the revised Table 3.

10. Table 3: This table does not provide new information, which is not already discussed in the
text. It also does not fit the format of a science publication to my opinion.

We have revised Table 3 to include only details of CO and $CH_4$ measurements.

[revised manuscript text omitted]

---

## Author Response (AR2)

**amt-2017-194**

**The CHRONOS mission: Capability for sub-hourly synoptic observations of carbon monoxide and methane to quantify emissions and transport of air pollution**

**David P. Edwards, Helen M. Worden, Doreen Neil, Gene Francis, Tim Valle, and Avelino F. Arellano Jr.**

We thank the reviewer for their careful evaluation of our manuscript. We address each comment (in blue) with an embedded response (in black) below. We detail new text that has been added to the revised manuscript (in green). Changes in the manuscript are also tracked (in green).

**Associate Editor Decision: Reconsider after major revisions** (27 Nov 2017) by Andre Butz
Comments to the Author:

Dear Dr. Edwards,
your manuscript underwent another independent review. Please carefully consider the comments by the referee for a revision.

In particular, I recommend reconsidering the "D/A argument". As noted by the referee, the difficulty with scattering radiative transfer is that the scattering lightpath is entirely different for absorbing and non-absorbing wavelengths if absorption optical depth is not small. So, a differential approach will not cancel the scattering effects (not even in first approximation) even though the particle single-scattering properties only depend weakly on wavelength.

Best regards,
André Butz.

The CO and CH4 gas absorption in the spectral regions of interest here does not approach optically thick. As we discuss in response to the Reviewer's question (referenced at l 378), the intent for these GEO observations is to only perform retrievals for pixels identified as cloud and aerosol clear. Our expected high-density sampling with sub-hourly observations permits this conservative strategy.

Though not necessarily appropriate for discussion in the paper, the CHRONOS concept described here was developed in response to the $100M cost-constrained mission criteria for the NASA Earth Venture Instrument (EVI) opportunity. This precluded our consideration of including additional measurements as part of the instrument concept. Although one can think of additional observations that might be useful, the supplementary meteorological and surface-property data that CHRONOS could use will be available through other GEO missions over North America such as GOES-16, TEMPO and geoCARB.

Report #1
Submitted on 27 Nov 2017
Anonymous Referee #3
Recommendation to the editor

1) Scientific significance

Does the manuscript represent a substantial contribution to scientific progress within the scope
of this journal (substantial new concepts, ideas, methods, or data)?

**Excellent**

2) Scientific quality

Are the scientific approaches and applied methods valid? Are the results discussed in an
appropriate and balanced way (consideration of related work, including appropriate
references)?

**Fair**

3) Presentation quality

Are the scientific results and conclusions presented in a clear, concise, and well structured way
(number and quality of figures/tables, appropriate use of English language)?

**Good**

For final publication, the manuscript should be

**accepted subject to minor revisions**

**Suggestions for revision or reasons for rejection (will be published if the paper is accepted for**
**final publication)**

The paper describes a new space mission and discusses its capabilities. As such the paper is
clearly within the scope of AMT. As far as I can judge, references to existing work are adequate.
I have just a number of issues which, I think, need consideration or further discussion.

l 59-62: The numbers presented here are estimates only but the text sounds as if they
represent the absolute truth. A weaker wording is necessary here, e.g., "are estimated to
decrease average life..."

We agree and have change the wording as suggested.

l 81: Is CHRONOS an acronym? If yes, please define; if not, why all capital letters?

CHRONOS was indeed originally an acronym. However, the original tile represented by the
letters is no longer applicable. At the same time, the name has gained recognition in the NASA

atmospheric composition community. We therefor chose to keep the name as is. CHRONOS
brings to mind time for the dense time-sampling of the observations.

Figure 3: I assume that the CHRONOS spatial resolution has been derived from the instrument
characteristics. However, the retrieval scheme is optimal estimation, and if the horizontal
resolution of the a priori information is limited, this would also degrade the horizontal
resolution of the result (even if L2 analysis is made pixel by pixel and no formal a priori
horizontal covariances are considered).

We understand the Reviewer's point that the instrument pixel size and retrieved product
resolution are not necessarily the same. The CHRONOS Level 2 product considers single-pixel
optimal estimation retrievals and no formal horizontal covariance is assumed as summized by
the Reviewer (the retrieval only considers vertical covariance). In as much as a radiation path
has a horizontal extent when taking into account the viewing and solar angles, a horizontal
covariance could be considered. However, at the same time, many operational retrievals
choose to use to globally constant a priori (e.g. IASI) to ensure that differences between
retrievals truly represent measured differences in gas concentration and not artifacts of a
variable a priori. Product versions for a given instrument often change retrieval a priori based
on experience with the measurements or newly available models/measurements to construct a
priori fields. We actually anticipate that direct radiance assimilation of the CHRONOS
measurements into a model will be the eventual path for most applications (but not all). The
analysis resolution would then be most relevant.

At Line 240 we have clarified that the grid shown in Fig. 3 is the pixel resolution, and at Line 604
that single pixel retrieval results depend on both the choice of a priori profile and a priori error
covariance.

Figure 4 is misleading because here the A-signal and the D-signal seem to be spectral radiances
while the text says these are band-integrated radiances. The information that the analysis is
based on band-integrated spectral radiances comes quite late (line 370) and the reader might
already be confused before reaching this line. For the authors and some readers from the gas
correlation filter radiometry community this is certainly clear but AMT articles are written for a
wider readership. Please remind the reader earlier in the paper that the information is
contained in the band-integrated radiances.

We have modified the caption in Figure 4 to indicate "spectrally-correlated, band-integrated
radiances" at Line 325, and the text has also been changed at Line 370 to indicate "spectrally-
correlated, band-integrated radiances".

l 378: It is not quite clear to me how the D/A approach solves the aerosol or cloud problem.  If
my understanding is correct, the D/A approach helps to remove the influence of the
background radiation in a way that the total column between the background source and TOA
is retrieved correctly. However, in the case of opaque clouds or aerosol layers, we get information about the partial column between the aerosol or cloud layer and TOA and not
about the total column between surface and TOA.  As far as I can judge, the D/A approach does
not remove the artefact arising from the misinterpretation of this partial column as total
column.  Or let me put it differently: The authors say that "to first order" their approach
reduces related uncertainties. "To first order" is often used as a synonym for "In linear
approximation" but radiative transfer in the case of opaque cloud or aerosol layers is extremely
non-linear. The capability of the D/A approach to solve related problems remains unclear. This
issue seems to me to deserve some more thorough discussion.

We agree with the Reviewer's statement. For cases of low AOD (e.g., figure 7 with AOD =
0.089), the effect of aerosol contamination is nearly cancelled in D/A, but this will not be the
case for scenes with optically thick aerosols or clouds. Data points are discarded when
identified as contaminated by opaque cloud or aerosol layers (for example, using the GOES ABI
cloud mask), and the D/A approach is not used in cases of opaque cloud or aerosol. CHRONOS
does not report retrievals in the presence of cloud or aerosol.  We have clarified this point with
additional text at Line 376:

", Therefore, the ratio of the D-signal and A-signal, D/A, eliminates the background
radiance term and reduces the impact of uncertainties associated with surface reflectance,
interfering gases, or optically thin aerosols and clouds.  In non-optically thin cases of clouds or
aerosols (OD $\gtrsim$ 0.2, identified using the GOES ABI cloud mask for example), data are discarded,
and no retrieval is performed. This approach is possible due to the high temporal and spatial
sampling of CHRONOS and the availability of ancillary cloud and aerosol geostationary
observations, both current and expected (e.g., Heidinger, 2011).

Changed ref:

Heidinger, A.: ABI Cloud Mask Algorithm Theoretical Basis Document, NOAA, 2011.
http://www.goes-r.gov/products/ATBDs/baseline/Cloud_CldMask_v2.0_no_color.pdf, last
access: 8 December 2017.

Text has also been added at Line 407 to indicate retrievals for optically thin aerosol and cloud
scenes.

l 447 Add blank before "As".

Corrected.

l 589, 597, 643: In Rodgers (2000) the Bayesian (or Shannonian) method is no longer called
"optimal estimation" but "maximum a posteriori". To my knowledge, Clive Rodgers has decided
no longer to use the term "optimal estimation" because there are many possible optimality
criteria.

We accept the Reviewer's point that Rogers's preference is for maximum a posteriori, although
common usage of optimal estimation persists. We now use maximum a posteriori consistently
throughout the manuscript.

Sect 5: The retrieval diagnostics strongly depend on how the constraint in the optimal
estimation retrieval is set up. How large are the a priori errors? Are the numbers used ad hoc
choices or can they be justified? Are there a priori covariances in the altitude domain or is the a
priori covariance matrix used diagonal? Is the abundance or the log of it retrieved? The text
provides, as far as I have seen, no information on these issues although the results critically
depend on them. Only in the legend and the caption of Fig 14. I found at least some information
for CH4; I have not found such information for CO. Without such information, the results are
not traceable. The results should be discussed in the context of these choices.

We agree with the reviewer that retrieval results and diagnostics have a strong dependence on
the retrieval constraints. We neglected to show this for the MOPITT results in Fig. 13 since this
information is described in numerous MOPITT publications, but we should have been more
explicit. We have rewritten the following paragraph in Sec. 5.1:

[revised manuscript text omitted]